# Deep Neural Networks Tend To Extrapolate Predictably

**Katie Kang**[1]**, Amrith Setlur**[2]**, Claire Tomlin**[1]**, Sergey Levine**[1]
[1]UC Berkeley, [2]Carnegie Mellon University

## Abstract

Conventional wisdom suggests that neural network predictions tend to be unpredictable and overconfident when faced with out-of-distribution (OOD) inputs. Our work reassesses this assumption for neural networks with high-dimensional inputs. Rather than extrapolating in arbitrary ways, we observe that neural network predictions often tend towards a constant value as input data becomes increasingly OOD. Moreover, we find that this value often closely approximates the optimal constant solution (OCS), i.e., the prediction that minimizes the average loss over the training data without observing the input. We present results showing this phenomenon across 8 datasets with different distributional shifts (including CIFAR10-C and ImageNet-R, S), different loss functions (cross entropy, MSE, and Gaussian NLL), and different architectures (CNNs and transformers). Furthermore, we present an explanation for this behavior, which we first validate empirically and then study theoretically in a simplified setting involving deep homogeneous networks with ReLU activations. Finally, we show how one can leverage our insights in practice to enable risk-sensitive decision-making in the presence of OOD inputs.

## 1 Introduction

The prevailing belief in machine learning posits that deep neural networks behave erratically when presented with out-of-distribution (OOD) inputs, often yielding predictions that are not only incorrect, but incorrect with *high confidence* (Guo et al., 2017; Nguyen et al., 2015). However, there is some evidence which seemingly contradicts this conventional wisdom – for example, Hendrycks & Gimpel (2016) show that the softmax probabilities outputted by neural network classifiers actually tend to be *less confident* on OOD inputs, making them surprisingly effective OOD detectors. In our work, we find that this softmax behavior may be reflective of a more general pattern in the way neural networks extrapolate: as inputs diverge further from the training distribution, a neural network's predictions often converge towards a *fixed* constant value. Moreover, this constant value often approximates the best prediction the network can produce without observing any inputs, which we refer to as the optimal constant solution (OCS). We call this the "reversion to the OCS" hypothesis:

> *Neural networks predictions on high-dimensional OOD inputs*
> *tend to revert towards the optimal constant solution.*

In classification, the OCS corresponds to the marginal distribution of the training labels, typically a high-entropy distribution. Therefore, our hypothesis posits that classifier outputs should become higher-entropy as the input distribution becomes more OOD, which is consistent with the findings in Hendrycks & Gimpel (2016). Beyond classification, to the best of our knowledge, we are the first to present and provide evidence for the "reversion to the OCS" hypothesis in its full generality. Our experiments show that the amount of distributional shift correlates strongly with the distance between model outputs and the OCS across 8 datasets, including both vision and NLP domains, 3 loss functions, and for both CNNS and transformers. We additionally report instances where this phenomenon did not hold, including for adversarial inputs.

Having made this observation, we set out to understand why neural networks have a tendency to behave this way. Our empirical analysis reveals that the feature representations corresponding to

---

Correspondence to: katiekang@eecs.berkeley.edu
Code: `https://github.com/katiekang1998/cautious_extrapolation`

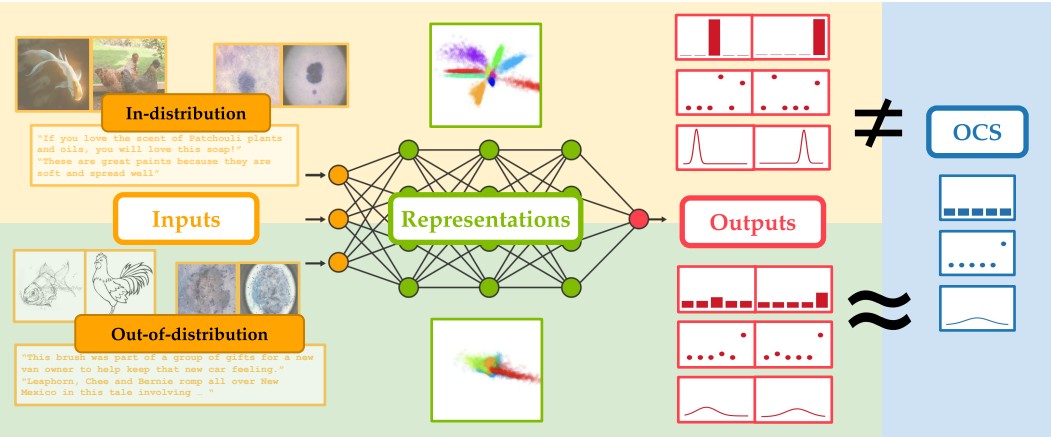

Figure 1: A summary of our findings. On in-distribution data (top), neural network outputs tend to vary significantly based on input labels. In contrast, on OOD data (bottom), we observe that model predictions tend to not only be more similar to one another, but also gravitate towards the optimal constant solution (OCS). We also observe that OOD inputs tend to map to representations with smaller magnitudes, leading to predictions largely dominated by the (constant) network biases, which may shed light on why neural networks have this tendency.

OOD inputs tend to have smaller norms than those of in-distribution inputs, leading to less signal being propagated from the input. As a result, neural network outputs from OOD inputs tend to be dominated by the input-independent parts of the network (e.g., bias vectors at each layer), which we observe to often map closely to the OCS. We also theoretically analyze the extrapolation behavior of deep homogeneous networks with ReLU activations, and derived evidence which supports this mechanism in the simplified setting.

Lastly, we leverage our observations to propose a simple strategy to enable risk-sensitive decision-making in the face of OOD inputs. The OCS can be viewed as a "backup default output" to which the neural network reverts when it encounters novel inputs. If we design the loss function such that the OCS aligns with the desirable cautious behavior as dictated by the decision-making problem, then the neural network model will automatically produce cautious decisions when its inputs are OOD. We describe a way to enable this alignment, and empirically demonstrate that this simple strategy can yield surprisingly good results in OOD selective classification.

## 2 RELATED WORK

A large body of prior works have studied various properties of neural network extrapolation. One line of work focuses on the failure modes of neural networks when presented with OOD inputs, such as poor generalization and overconfidence (Torralba & Efros, 2011; Gulrajani & Lopez-Paz, 2020; Recht et al., 2019; Ben-David et al., 2006; Koh et al., 2021). Other works have noted that neural networks are ineffective in capturing epistemic uncertainty in their predictions (Ovadia et al., 2019; Lakshminarayanan et al., 2017; Nalisnick et al., 2018; Guo et al., 2017; Gal & Ghahramani, 2016), and that a number of techniques can manipulate neural networks to produce incorrect predictions with high confidence (Szegedy et al., 2013; Nguyen et al., 2015; Papernot et al., 2016; Hein et al., 2019). However, Hendrycks et al. (2018) observed that neural networks assign lower maximum softmax probabilities to OOD than to in-distribution point, meaning neural networks may actually exhibit less confidence on OOD inputs. Our work supports this observation, while further generalizing it to arbitrary loss functions. Other lines of research have explored OOD detection via the norm of the learned features (Sun et al., 2022; Tack et al., 2020), the influence of architectural decisions on generalization (Xu et al., 2020; Yehudai et al., 2021; Cohen-Karlik et al., 2022; Wu et al., 2022), the relationship between in-distribution and OOD performance (Miller et al., 2021; Baek et al., 2022; Balestriero et al., 2021), and the behavior of neural network representations under OOD conditions (Webb et al., 2020; Idnani et al., 2022; Pearce et al., 2021; Dieterich & Guyer, 2022; Huang et al., 2020). While our work also analyzes representations in the context of extrapolation, our focus is on understanding the mechanism behind "reversion to the OCS", which differs from the aforementioned works.

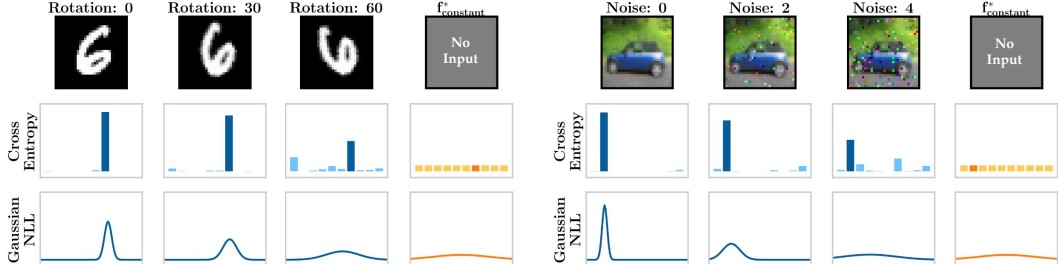

Figure 2: Neural network predictions from training with cross entropy and Gaussian NLL on MNIST (left) and CIFAR10 (right). The models were trained with 0 rotation/noise, and evaluated on increasingly OOD inputs consisting of the digit 6 for MNIST, and of automobiles for CIFAR10. The blue plots represent the average model prediction over the evaluation dataset. The orange plots show the OCS associated with each model. We can see that as the distribution shift increases, the network predictions tend towards the OCS.

Our work also explores risk-sensitive decision-making using selective classification as a testbed. Selective classification is a well-studied problem, and various methods have been proposed to enhance selective classification performance (Geifman & El-Yaniv, 2017; Feng et al., 2011; Charoenphakdee et al., 2021; Ni et al., 2019; Cortes et al., 2016; Xia & Bouganis, 2022). In contrast, our aim is not to develop the best possible selective classification approach, but rather to providing insights into the effective utilization of neural network predictions in OOD decision-making.

## 3 REVERSION TO THE OPTIMAL CONSTANT SOLUTION

In this work, we will focus on the widely studied covariate shift setting (Gretton et al., 2009; Sugiyama et al., 2007). Formally, let the training data $\mathcal{D} = \{(x_i, y_i)\}_{i=1}^N$ be generated by sampling $x_i \sim P_{\text{train}}(x)$ and $y_i \sim P(y|x_i)$. At test time, we query the model with inputs generated from $P_{\text{OOD}}(x) \neq P_{\text{train}}(x)$, whose ground truth labels are generated from the same conditional distribution, $P(y|x)$, as that in training. We will denote a neural network model as $f_\theta : \mathbb{R}^d \to \mathbb{R}^m$, where $d$ and $m$ are the dimensionalities of the input and output, and $\theta \in \Theta$ represents the network weights. We will focus on settings where $d$ is high-dimensional. The neural network weights are optimized by minimizing a loss function $\mathcal{L}$ using gradient descent, $\hat{\theta} = \arg\min_{\theta \in \Theta} \frac{1}{N} \sum_{i=1}^N \mathcal{L}(f_\theta(x_i), y_i)$.

**Main hypothesis.** In our experiments, we observed that as inputs become more OOD, neural network predictions tend to revert towards a constant prediction. This means that, assuming there is little label shift, model predictions will tend to be more similar to one another for OOD inputs than for the training distribution. Furthermore, we find that this constant prediction is often similar to the optimal constant solution (OCS), which minimizes the training loss if the network is constrained to ignore the input. More precisely, we define the OCS as $f_{\text{constant}}^* = \arg\min_{f \in \mathbb{R}^m} \frac{1}{N} \sum_{1 \leq i \leq N} \mathcal{L}(f, y_i)$. Based on our observations, we hypothesize that **as the likelihood of samples from $P_{\text{OOD}}(x)$ under $P_{\text{train}}(x)$ decreases, $f_{\hat{\theta}}(x)$ for $x \sim P_{\text{OOD}}(x)$ tends to approach $f_{\text{constant}}^*$.** As an illustrative example, we trained models using either cross-entropy or (continuous-valued) Gaussian negative log-likelihood (NLL) on the MNIST and CIFAR10 datasets. The blue plots in Fig. 2 show the models' predictions as its inputs become increasingly OOD, and the orange plots visualize the OCS associated with each model. We can see that even though we trained on *different* datasets and evaluated on *different* kinds of distribution shifts, the neural network predictions exhibit the *same* pattern of extrapolation: as the distribution shift increases, the network predictions move closer to the OCS.

### 3.1 EXPERIMENTS

We will now provide empirical evidence for the "reversion to the OCS" hypothesis. Our experiments aim to answer the question: **As the test-time inputs become more OOD, do neural network predictions move closer to the optimal constant solution?**

**Experimental setup.** We trained our models on 8 different datasets, and evaluated them on both natural and synthetic distribution shifts: CIFAR10/CIFAR10-C, ImageNet / ImageNet-R(endition) and Sketch, DomainBed OfficeHome, SkinLesionPixels, UTKFace, BREEDS living-17, BREEDS non-living-26, and WILDS Amazon. See Appendix B.1 for a more detailed description of each dataset. Models with image inputs use ResNet (He et al., 2016) or VGG (Simonyan & Zisserman,

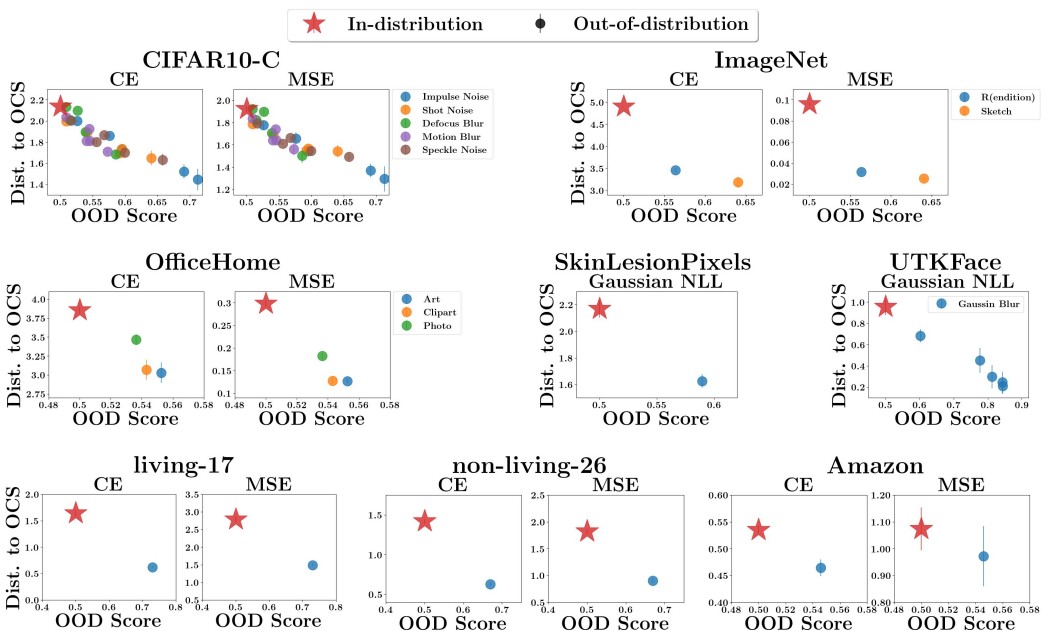

Figure 3: Evaluating the distance between network predictions and the OCS as the input distribution becomes more OOD. Each point represents a different evaluation dataset, with the red star representing the (holdout) training distribution, and circles representing OOD datasets. The vertical line associated with each point represents the standard deviation over 5 training runs. As the OOD score of the evaluation dataset increases, there is a clear trend of the neural network predictions approaching the OCS.

2014) style architectures, and models with text inputs use DistilBERT (Sanh et al., 2019), a distilled version of BERT (Devlin et al., 2018).

We focus on three tasks, each using a different loss functions: classification with cross entropy (CE), selective classification with mean squared error (MSE), and regression with Gaussian NLL. Datasets with discrete labels are used for classification and selective classification, and datasets with continuous labels are used for regression. The cross entropy models are trained to predict the likelihood that the input belongs to each class, as is typical in classification. The MSE models are trained to predict rewards for a selective classification task. More specifically, the models output a value for each class as well as an abstain option, where the value represents the reward of selecting that option given the input. The ground truth reward is +1 for the correct class, -4 for the incorrect classes, and +0 for abstaining. We train these models by minimizing the MSE loss between the predicted and ground truth rewards. We will later use these models for decision-making in Section 5. The Gaussian NLL models predict a mean and a standard deviation, parameterizing a Gaussian distribution. They are trained to minimize the negative log likelihood of the labels under its predicted distributions.

**Evaluation protocol.** To answer our question, we need to quantify (1) the dissimilarity between the training data and the evaluation data, and (2) the proximity of network predictions to the OCS. To estimate the former, we trained a low-capacity model to discriminate between the training and evaluation datasets and measured the average predicted likelihood that the evaluation dataset is generated from the evaluation distribution, which we refer to as the OOD score. This score is $0.5$ for indistinguishable train and evaluation data, and $1$ for a perfect discriminator. To estimate the distance between the model's prediction and the OCS, we compute the KL divergence between the model's predicted distribution and the distribution parameterized by the OCS, $\frac{1}{N} \sum_{i=1}^{N} D_{\mathrm{KL}}(P_\theta(y|x_i) || P_{f^*_{\mathrm{constant}}}(y))$, for models trained with cross-entropy and Gaussian NLL. For MSE models, the distance is measured using the mean squared error, $\frac{1}{N} \sum_{i=1}^{N} ||f_\theta(x_i) - f^*_{\mathrm{constant}}||^2$. See Appendix B.3 for more details on our evaluation protocol, and Appendix B.4 for the closed form solution for the OCS for each loss.

**Results.** In Fig. 3, we plot the OOD score (x-axis) against the distance between the network predictions and the OCS (y-axis) for both the training and OOD datasets. Our results indicate a clear trend: as the OOD score of the evaluation dataset increases, neural network predictions move closer

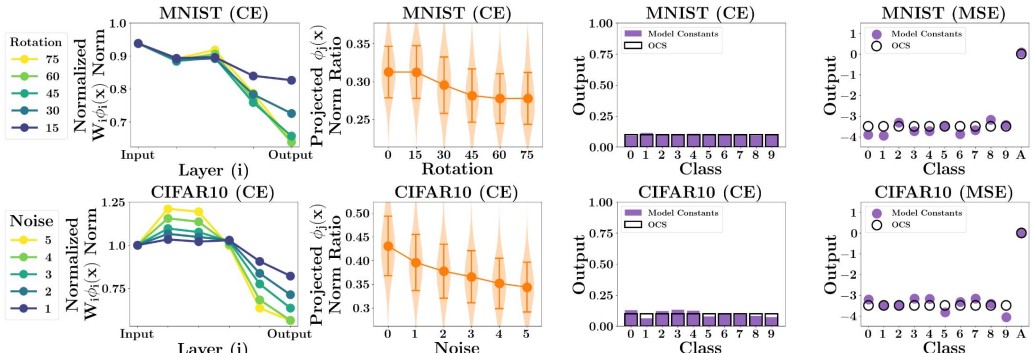

Figure 4: Analysis of the interaction between representations and weights as distribution shift increases. Plots in first column visualize the norm of network features for different levels of distribution shift at different layers of the network. In later layer of the network, the norm of features tends to decrease as distribution shift increases. Plots in second column show the proportion of network features which lie within the span of the following linear layer. This tends to decrease as distributional shift increases. Error bars represent the standard deviation taken over the test distribution. Plots in the third and fourth column show the accumulation of model constants as compared to the OCS for a cross entropy and a MSE model; the two closely mirror one another.

to the OCS. Moreover, our results show that this trend holds relatively consistently across different loss functions, input modalities, network architectures, and types of distribution shifts. We also found instances where this phenomenon did not hold, such as adversarial inputs, which we discuss in greater detail in Appendix A. However, the overall prevalence of "reversion to the OCS" across different settings suggests that it may capture a general pattern in the way neural networks extrapolate.

## 4    WHY DO OOD PREDICTIONS REVERT TO THE OCS?

In this section, we aim to provide insights into why neural networks have a tendency to revert to the OCS. We will begin with an intuitive explanation, and provide empirical and theoretical evidence in Sections 4.1 and 4.2. In our analysis, we observe that weight matrices and network representations associated with training inputs often occupy low-dimensional subspaces with high overlap. However, when the network encounters OOD inputs, we observe that their associated representations tend to have less overlap with the weight matrices compared to those from the training distribution, particularly in the later layers. As a result, OOD representations tend to diminish in magnitude as they pass through the layers of the network, causing the network's output to be primarily influenced by the accumulation of model constants (e.g. bias terms). Furthermore, both empirically and theoretically, we find that this accumulation of model constants tend to closely approximate the OCS. We posit that reversion to the OCS for OOD inputs occurs due to the combination of these two factors: that accumulated model constants in a trained network tend towards the OCS, and that OOD points yield smaller-magnitude representations in the network that become dominated by model constants.

### 4.1    EMPIRICAL ANALYSIS

We will now provide empirical evidence for the mechanism we describe above using deep neural network models trained on MNIST and CIFAR10. MNIST models use a small 4 layer network, and CIFAR10 models use a ResNet20 (He et al., 2016). To more precisely describe the quantities we will be illustrating, let us rewrite the neural network as $f(x) = g_{i+1}(\sigma(W_i\phi_i(x) + b_i))$, where $\phi_i(x)$ is an intermediate representation at layer $i$, $W_i$ and $b_i$ are the corresponding weight matrix and bias, $\sigma$ is a nonlinearity, and $g_{i+1}$ denotes the remaining layers of the network. Because we use a different network architecture for each domain, we will use variables to denote different intermediate layers of the network, and defer details about the specific choice of layers to Appendix C.2. We present additional experiments analyzing the ImageNet domain, as well as the effects of batch and layer normalization in Appendix C.1.

First, we will show that $W_i\phi_i(x)$ tends to diminish for OOD inputs. The first column of plots in Fig. 4 show $\mathbb{E}_{x \sim P_{\text{OOD}}(x)}[||W_i\phi_i(x)||^2]/\mathbb{E}_{x \sim P_{\text{train}}(x)}[||W_i\phi_i(x)||^2]$ for $P_{\text{OOD}}$ with different level of rotation or noise. The x-axis represents different layers in the network, with the leftmost being the input

and the rightmost being the output. We can see that in the later layers of the network, $||W_i\phi_i(x)||^2$ consistently became smaller as inputs became more OOD (greater rotation/noise). Furthermore, the diminishing effect becomes more pronounced as the representations pass through more layers.

Next, we will present evidence that this decrease in representation magnitude occurs because $\phi_j(x)$ for $x \sim P_{\text{train}}(x)$ tend to lie more within the low-dimensional subspace spanned by the rows of $W_j$ than $\phi_j(x)$ for $x \sim P_{\text{OOD}}(x)$. Let $V_{\text{top}}$ denote the top (right) singular vectors of $W_j$. The middle plots of Fig. 4 show the ratio of the representation's norm at layer $j$ that is captured by projecting the representation onto $V_{\text{top}}$ i.e., $||\phi_j(x)^\top V_{\text{top}} V_{\text{top}}^\top||^2 / ||\phi_j(x)||^2$, as distribution shift increases. We can see that as the inputs become more OOD, the ratio goes down, suggesting that the representations lie increasingly outside the subspace spanned by the weight matrix.

Finally, we will provide evidence for the part of the mechanism which accounts for the optimality of the OCS. Previously, we have established that OOD representations tend to diminish in magnitude in later layers of the network. This begs the question, what would the output of the network be if the input representation at an intermediary layer had a magnitude of 0? We call this the accumulation of model constants, i.e. $g_{k+1}(\sigma(b_k))$. In the third and fourth columns of Fig. 4, we visualize the accumulation of model constants at one of the final layers $k$ of the networks for both a cross entropy and a MSE model (details in Sec. 3.1), along with the OCS for each model. We can see that the accumulation of model constants closely approximates the OCS in each case.

## 4.2 THEORETICAL ANALYSIS

We will now explicate our empirical findings more formally by analyzing solutions of gradient flow (gradient descent with infinitesimally small step size) on deep homogeneous neural networks with ReLU activations. We adopt this setting due to its theoretical convenience in reasoning about solutions at convergence of gradient descent (Lyu & Li, 2019; Galanti et al., 2022; Huh et al., 2021), and its relative similarity to deep neural networks used in practice (Neyshabur et al., 2015; Du et al., 2018).

**Setup:** We consider a class of homogeneous neural networks $\mathcal{F} := \{f(W; x) : W \in \mathcal{W}\}$, with $L$ layers and ReLU activation, taking the functional form $f(W; x) = W_L \sigma(W_{L-1} \ldots \sigma(W_2 \sigma(W_1 x)) \ldots)$, where $W_i \in \mathbb{R}^{m \times m}, \forall i \in \{2, \ldots, L-1\}$, $W_1 \in \mathbb{R}^{m \times 1}$ and $W_L \in \mathbb{R}^{1 \times m}$. Our focus is on a binary classification problem where we consider two joint distributions $P_{\text{train}}, P_{\text{OOD}}$ over inputs and labels: $\mathcal{X} \times \mathcal{Y}$, where inputs are from $\mathcal{X} := \{x \in \mathbb{R}^d : \|x\|_2 \leq 1\}$, and labels are in $\mathcal{Y} := \{-1, +1\}$. We consider gradient descent with a small learning rate on the objective: $L(W; \mathcal{D}) := \sum_{(x,y) \in \mathcal{D}} \ell(f(W; x), y)$ where $\ell(f(W; x), y) \mapsto \exp(-yf(W; x))$ is the exponential loss and $\mathcal{D}$ is an IID sampled dataset of size $N$ from $P_{\text{train}}$. For more details on the setup, background on homogeneous networks, and full proofs for all results in this section, please see Appendix D.

We will begin by providing a lower bound on the expected magnitude of intermediate layer features corresponding to inputs from the training distribution:

**Proposition 4.1** ($P_{\text{train}}$ **observes high norm features**) *When $f(\hat{W}; x)$ fits $\mathcal{D}$, i.e., $y_i f(\hat{W}; x_i) \geq \gamma$, $\forall i \in [N]$, then w.h.p $1 - \delta$ over $\mathcal{D}$, layer $j$ representations $f_j(\hat{W}; x)$ satisfy $\mathbb{E}_{P_{\text{train}}}[\|f_j(\hat{W}; x)\|_2] \geq (1/C_0)(\gamma - \tilde{\mathcal{O}}(\sqrt{\log(1/\delta)/N} + C_1 \log m / N\gamma))$, if $\exists$ constants $C_0, C_1$ s.t. $\|\hat{W}_j\|_2 \leq C_0^{1/L}$, $C_1 \geq C_0^{3L/2}$.*

Here, we can see that if the trained network perfectly fits the training data ($y_i f(\hat{W}; x_i) \geq \gamma$, $\forall i \in [N]$), and the training data size $N$ is sufficiently large, then the expected $\ell_2$ norm of layer $j$ activations $f_j(\hat{W}; x)$ on $P_{\text{train}}$ is large and scales at least linearly with $\gamma$.

Next, we will analyze the size of network outputs corresponding to points which lie outside of the training distribution. Our analysis builds on prior results for gradient flow on deep homogeneous nets with ReLU activations which show that the gradient flow is biased towards the solution (KKT point) of a specific optimization problem: minimizing the weight norms while achieving sufficiently large margin on each training point (Timor et al., 2023; Arora et al., 2019; Lyu & Li, 2019). Based on this, it is easy to show that that the solution for this constrained optimization problem is given by a neural network with low rank matrices in each layer for sufficiently deep and wide networks. Furthermore, the low rank nature of these solutions is exacerbated by increasing depth and width, where the network approaches an almost rank one solution for each layer. If test samples deviate from this low rank space of weights in any layer, the dot products of the weights and features will collapse in the subsequent layer, and its affect rolls over to the final layer, which will output features

with very small magnitude. Using this insight, we present an upper bound on the magnitude of the final layer features corresponding to OOD inputs:

**Theorem 4.1 (Feature norms can drop easily on $P_{\text{OOD}}$)** *Suppose $\exists$ a network $f'(W; x)$ with $L'$ layers and $m'$ neurons satisfying conditions in Proposition 4.1 ($\gamma$=1). When we optimize the training objective with gradient flow over a class of deeper and wider homogeneous networks $\mathcal{F}$ with $L > L', m > m'$, the resulting solution would converge directionally to a network $f(\hat{W}; x)$ for which the following is true: $\exists$ a set of rank 1 projection matrices $\{A_i\}_{i=1}^{L}$, such that if representations for any layer $j$ satisfy $\mathbb{E}_{P_{\text{OOD}}}\|A_j f_j(\hat{W}; x)\|_2 \leq \epsilon$, then $\exists C_2$ for which $\mathbb{E}_{P_{\text{OOD}}}[|f(\hat{W}; x)|] \lesssim C_0(\epsilon + C_2^{-1/L}\sqrt{L+1/L})$.*

This theorem tells us that for any layer $j$, there exists only a narrow rank one space $A_j$ in which OOD representations may lie, in order for their corresponding final layer outputs to remain significant in norm. Because neural networks are not optimized on OOD inputs, we hypothesize that the features corresponding to OOD inputs tend to lie outside this narrow space, leading to a collapse in last layer magnitudes for OOD inputs in deep networks. Indeed, this result is consistent with our empirical findings in the first and second columns of Fig. 4.1, where we observed that OOD features tend to align less with weight matrices, resulting in a drop in OOD feature norms.

To study the accumulation of model constants, we now analyze a slightly modified class of functions $\tilde{\mathcal{F}} = \{f(W; \cdot) + b : b \in \mathbb{R}, f(W; \cdot) \in \mathcal{F}\}$, which consists of deep homogeneous networks with a bias term in the final layer. In Proposition 4.2, we show that there exists a set of margin points (analogous to support vectors in the linear setting) which solely determines the model's bias $\hat{b}$.

**Proposition 4.2 (Analyzing network bias)** *If gradient flow on $\tilde{\mathcal{F}}$ converges directionally to $\hat{W}, \hat{b}$, then $\hat{b} \propto \sum_k y_k$ for margin points $\{(x_k, y_k) : y_k \cdot f(\hat{W}; x_k) = \arg\min_{j \in [N]} y_j \cdot f(\hat{W}; x_j)\}$.*

If the label marginal of these margin points mimics that of the overall training distribution, then the learnt bias will approximate the OCS for the exponential loss. This result is consistent with our empirical findings in the third and fourth columns on Fig. 4.1, where we found the accumulation of bias terms tends to approximate the OCS.

## 5 RISK-SENSITIVE DECISION-MAKING

Lastly, we will explore an application of our observations to decision-making problems. In many decision-making scenarios, certain actions offer a high potential for reward when the agent chooses them correctly, but also higher penalties when chosen incorrectly, while other more cautious actions consistently provide a moderate level of reward. When utilizing a learned model for decision-making, it is desirable for the agent to select the high-risk high-reward actions when the model is likely to be accurate, while opting for more cautious actions when the model is prone to errors, such as when the inputs are OOD. It turns out, if we leverage "reversion to the OCS" appropriately, such risk-sensitive behavior can emerge automatically. If the OCS of the agent's learned model corresponds to cautious actions, then "reversion to the OCS" posits that the agent will take increasingly cautious actions as its inputs become more OOD. However, not all decision-making algorithms leverage "reversion to the OCS" by default. **Depending on the choice of loss function (and consequently the OCS), different algorithms which have similar in-distribution performance can have different OOD behavior.** In the following sections, we will use selective classification as an example of a decision-making problem to more concretely illustrate this idea.

### 5.1 EXAMPLE APPLICATION: SELECTIVE CLASSIFICATION

In selective classification, the agent can choose to classify the input or abstain from making a decision. As an example, we will consider a selective classification task using CIFAR10, where the agent receives a reward of +1 for correctly selecting a class, a reward of -4 for an incorrect classification, and a reward of 0 for choosing to abstain.

Let us consider one approach that leverages "reversion to the OCS" and one that does not, and discuss their respective OOD behavior. An example of the former involves learning a model to predict the reward associated with taking each action, and selecting the action with the highest

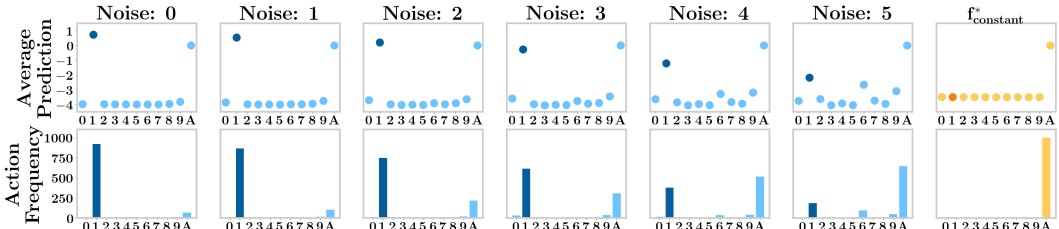

Figure 5: Selective classification via reward prediction on CIFAR10. We evaluate on holdout datasets consisting of automobiles (class 1) with increasing levels of noise. X-axis represents the agent's actions, where classes are indexed by numbers and abstain is represented by "A". We plot the average reward predicted by the model for each class (top), and the distribution of actions selected by the policy (bottom). The rightmost plots represent the OCS (top), and the actions selected by an OCS policy (bottom). As distribution shift increased, the model predictions approached the OCS, and the policy automatically selected the abstain action more frequently.

predicted reward. This reward model, $f_\theta : \mathbb{R}^d \to \mathbb{R}^{|\mathcal{A}|}$, takes as input an image, and outputs a vector the size of the action space. We train $f_\theta$ using a dataset of images, actions and rewards, $\mathcal{D} = \{(x_i, a_i, r_i)\}_{i=1}^N$, by minimizing the MSE loss, $\frac{1}{N} \sum_{1 \le i \le N} (f_\theta(x_i)_{a_i} - r_i)^2$, and select actions using the policy $\pi(x) = \arg\max_{a \in \mathcal{A}} f_\theta(x)_a$. The OCS of $f_\theta$ is the average reward for each action over the training points, i.e., $(f_{\text{constant}}^*)_a = \frac{\sum_{1 \le i \le N} r_i \cdot \mathbb{1}[a_i = a]}{\sum_{1 \le j \le N} \mathbb{1}[a_j = a]}$. In our example, the OCS is -3.5 for selecting each class and 0 for abstaining, so the policy corresponding to the OCS will choose to abstain. Thus, according to "reversion to the OCS", this agent should choose to abstain more and more frequently as its input becomes more OOD. We illustrate this behavior in Figure 5. In the first row, we depict the average predictions of a reward model when presented with test images of a specific class with increasing levels of noise (visualized in Figure 1). In the second row, we plot a histogram of the agent's selected actions for each input distribution. We can see that as the inputs become more OOD, the model's predictions converged towards the OCS, and consequently, the agent automatically transitioned from making high-risk, high-reward decisions of class prediction to the more cautious decision of abstaining.

One example of an approach which does not leverage "reversion to the OCS" is standard classification via cross entropy. The classification model takes as input an image and directly predicts a probability distribution over whether each action is the optimal action. In this case, the optimal action given an input is always its ground truth class. Because the OCS for cross entropy is the marginal distribution of labels in the training data, and the optimal action is never to abstain, the OCS for this approach corresponds to a policy that *never* chooses to abstain. In this case, "reversion to the OCS" posits that the agent will continue to make high-risk high-reward decisions even as its inputs become more OOD. As a result, while this approach can yield high rewards on the training distribution, it is likely to yield very low rewards on OOD inputs, where the model's predictions are likely to be incorrect.

## 5.2 Experiments

We will now more thoroughly compare the behavior of a reward prediction agent with a standard classification agent for selective classification on a variety of different datasets. Our experiments aim to answer the questions: **How does the performance of a decision-making approach which leverages "reversion to the OCS" compare to that of an approach which does not?**

**Experimental Setup.** Using the same problem setting as the previous section, we consider a selective classification task in which the agent receives a reward of +1 for selecting the correct class, -4 for selecting an incorrect class, and +0 for abstaining from classifying. We experiment with 4 datasets: CIFAR10, DomainBed OfficeHome, BREEDS living-17 and non-living-26. We compare the performance of the reward prediction and standard classification approaches described in the previous section, as well as a third oracle approach that is optimally risk-sensitive, thereby providing an upper bound on the agent's achievable reward. To obtain the oracle policy, we train a classifier on the training dataset to predict the likelihood of each class, and then calibrate the predictions with temperature scaling on the OOD evaluation dataset. We then use the reward function to calculate the theoretically optimal threshold on the classifier's maximum predicted likelihood, below which the

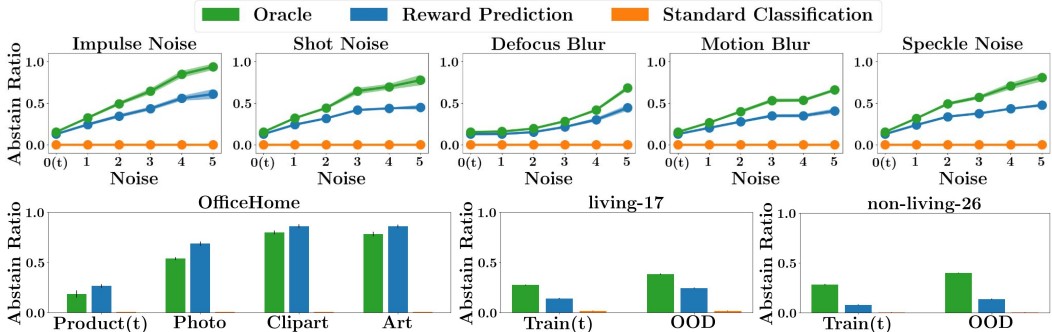

Figure 6: Ratio of abstain action to total actions; error bars represent standard deviation over 5 random seeds; (t) denotes the training distribution. While the oracle and reward prediction approaches selected the abstain action more frequently as inputs became more OOD, the standard classification approach almost never selected abstain.

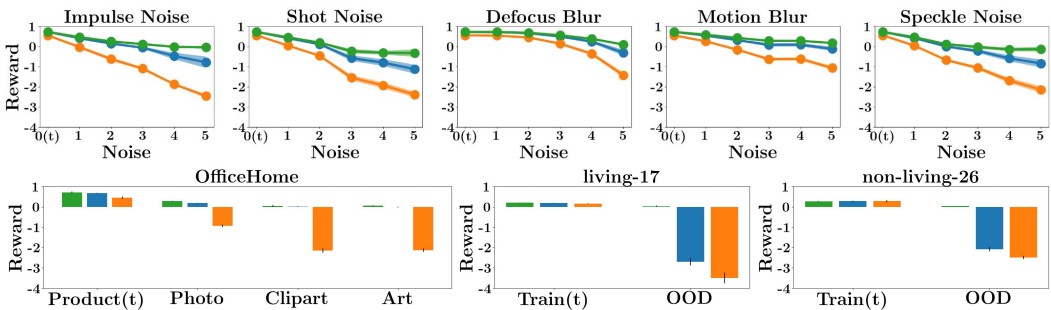

Figure 7: Reward obtained by each approach. While all three approaches performed similarly on the training distribution, reward prediction increasingly outperformed standard classification as inputs became more OOD.

abstaining option is selected. Note that the oracle policy has access to the reward function and the OOD evaluation dataset for calibration, which the other two approaches do not have access to.

**Results.** In Fig. 6, we plot the frequency with which the abstain action is selected for each approach. As distribution shift increased, both the reward prediction and oracle approaches selected the abstaining action more frequently, whereas the standard classification approach never selected this option. This discrepancy arises because the OCS of the reward prediction approach aligns with the abstain action, whereas the OCS of standard classification does not. In Fig. 7, we plot the average reward received by each approach. Although the performance of all three approaches are relatively similar on the training distribution, the reward prediction policy increasingly outperformed the classification policy as distribution shift increased. Furthermore, the gaps between the rewards yielded by the reward prediction and classification policies are substantial compared to the gaps between the reward prediction and the oracle policies, suggesting that the former difference in performance is nontrivial. Note that the goal of our experiments was not to demonstrate that our approach is the best possible method for selective classification (in fact, our method is likely not better than SOTA approaches), but rather to highlight how the OCS associated with an agent's learned model can influence its OOD decision-making behavior. To this end, this result shows that appropriately leveraging "reversion to the OCS" can substantial improve an agent's performance on OOD inputs.

## 6 CONCLUSION

We presented the observation that neural network predictions for OOD inputs tend to converge towards the model's optimal constant solution, and provided empirical evidence for this phenomenon across diverse datasets and different types of distributional shifts. We also proposed a mechanism to explain this phenomenon and a simple strategy that leverages this phenomenon to enable risk-sensitive decision-making. While our understanding of "reversion to the OCS" remains rudimentary, we hope our observations will prompt further investigations on how we may predict and even potentially steer the behavior of neural networks on OOD inputs.

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

# A INSTANCES WHERE "REVERSION TO THE OCS" DOES NOT HOLD

In this section, we will discuss some instances where "reversion to the OCS" does not hold. The first example is evaluating an MNIST classifier (trained via cross entropy) on an adversarially generated dataset using the Fast Gradient Sign Method (FSGM) (Goodfellow et al., 2014). In Fig. 8, we show our findings. On the left plot, we can see that the outputs corresponding to the adversarial dataset were farther from the OCS than the predictions corresponding to the training distribution, even though the adversarial dataset is more OOD. On the right plot, we show the normalized representation magnitude of the original and adversarial distributions throughout different layers of the network. We can see that the representations corresponding to adversarial inputs have larger magnitudes compared to those corresponding to the original inputs. This is a departure from the leftmost plots in Fig. 4, where the norm of the representations in later layers decreased as the inputs became more OOD. One potential reason for this behavior is that the adversarial optimization pushes the adversarial inputs to yield representations which align closer with the network weights, leading to higher magnitude representations which push outputs father from the OCS.

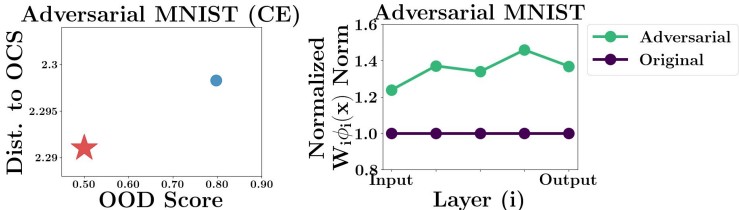

Figure 8: On the left, we evaluate the distance between network predictions and the OCS as the input distribution becomes more OOD for an MNIST classifier. The red star represents the (holdout) training distribution, and the blue circle represents an adversarially generated evaluation distribution. Even though the adversarial distribution is more OOD, its predictions were farther from the OCS. On the right, we plot the normalized representation magnitude across different layers of the network. The representations corresponding to adversarial inputs had greater magnitude throughout all the layers.

Our second example is Gaussian NLL models trained on UTKFace, and evaluated on inputs with impulse noise. Previously, in Fig. 3, we had shown that "reversion to the OCS" holds for UTKFace with gaussian blur, but we found this to not necessarily be the case for all corruptions. In Fig. 9, we show the behavior of the models evaluated on inputs with increasing amounts of impulse noise. In the middle plot, we can see that as the OOD score increases (greater noising), the distance to the OCS increases, contradicting "reversion to the OCS". In the right plot, we show the magnitude of the representations in an internal layer for inputs with both gaussian blur and impulse noise. We can see that while the representation norm decreases with greater noise for gaussian blur, the representation norm actually increases for impulse noise. We are not sure why the model follows "reversion to the OCS" for gaussian blur but not impulse noise. We hypothesize that one potential reason could be that, because the model was trained to predict age, the model learned to identify uneven texture as a proxy for wrinkles. Indeed, we found that the model predicted higher ages for inputs with greater levels of impulse noise, which caused the predictions to move farther from the OCS.

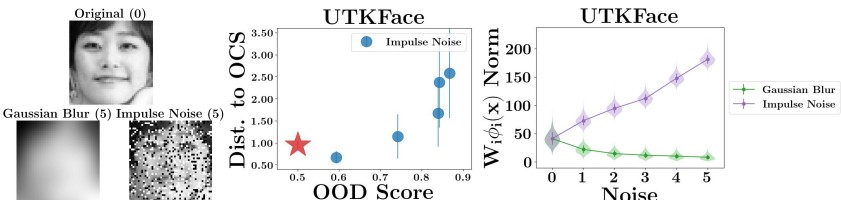

Figure 9: On the left, we visualize an example of UTKFace inputs in its original form, with gaussian blur, and with impulse noise. In the middle, we evaluate the distance between network predictions and the OCS as the input distribution becomes more OOD. Each point represents a different evaluation dataset, with the red star representing the (holdout) training distribution, and circles representing OOD datasets with increasing levels of impulse noise. The vertical line associated with each point represents the standard deviation over 5 training runs. As the OOD score of the evaluation dataset increases, the model predictions here tended to move farther from the OCS. On the right, we plot the magnitude of the representation in a specific layer of the model corresponding to inputs with different levels of gaussian blur and impulse noise. The representation magnitude for inputs with greater gaussian blur tend to decrease, while the representation magnitude with greater impulse noise tend to increase.

## B EXPERIMENT DETAILS

### B.1 DATASETS

The datasets with discrete labels include CIFAR10 (Krizhevsky et al., 2009), ImageNet (Deng et al., 2009) (subsampled to 200 classes to match ImageNet-R(rendition) (Hendrycks et al., 2021)), DomainBed OfficeHome (Gulrajani & Lopez-Paz, 2020), BREEDS LIVING-17 and NON-LIVING-26 (Santurkar et al., 2020), and Wilds Amazon (Koh et al., 2021), and the datasets with continuous labels include SkinLesionPixels (Gustafsson et al., 2023) and UTKFace (Zhang et al., 2017). We evaluate CIFAR10 models on CIFAR10-C (Hendrycks & Dietterich, 2019), which includes synthetic corruptions at varying levels of intensity. We evaluate ImageNet models on ImageNet-R and ImageNet-Sketch (Wang et al., 2019), which include renditions of ImageNet classes in novel styles and in sketch form. OfficeHome includes images of furniture in the style of product, photo, clipart, and art. We train on product images and evaluate on the other styles. BREEDS datasets consist of training and OOD test datasets consisting of the same classes but distinct subclasses. Wilds Amazon consists of training and OOD test datasets of Amazon reviews from different sets of users. To amplify the distribution shift, we subsample the OOD dataset to only include incorrectly classified points. SkinLesionPixels consists of dermatoscopic images, where the training and OOD test datasets were collected from patients from different countries. UTKFace consists of images of faces, and we construct OOD test datasets by adding different levels of Gaussian blur to the input.

### B.2 TRAINING PARAMETERS

First, we will discuss the parameters we used to train our models.

| Task | Network Architecture |
|---|---|
| | 2 convolution layers followed by 2 fully connected layers |
| MNIST | ReLU nonlinearities |
| CIFAR10 | ResNet20 |
| ImageNet | ResNet50 |
| OfficeHome | ResNet50 |
| BREEDS | ResNet18 |
| Amazon | DistilBERT |
| SkinLesionPixels | ResNet34 |
| UTKFace | Custom VGG style architecture |

| Task | Optimizer | Learning Rate | Learning Rate Scheduler | Weight Decay | Momentum |
|---|---|---|---|---|---|
| MNIST | Adam | 0.001 | Step; $\gamma = 0.7$ | 0.01 | - |
| CIFAR10 | SGD | 0.1 | Multi step milestones=[100, 150] | 0.0001 | 0.9 |
| ImageNet | SGD | 0.1 | Step; $\gamma = 0.1$ step size = 30 | 0.0001 | 0.9 |
| OfficeHome | Adam | 0.00005 | - | 0 | - |
| BREEDS | SGD | 0.2 | Linear with warm up (warm up frac=0.05) | 0.00005 | 0.9 |
| Amazon | AdamW | 0.00001 | Linear with warm up (warm up frac=0) | 0.01 | - |
| SkinLesionPixels | Adam | 0.001 | - | 0 | - |
| UTKFace | Adam | 0.001 | - | 0 | - |

| Task | Data Preprocessing |
|---|---|
| MNIST | Normalization |
| CIFAR10 | Random horizontal flip, random crop, normalization |
| ImageNet | Random resized crop, random horizontal flip, normalization |
| OfficeHome | Random resized crop, random horizontal flip, color jitter, random grayscale, normalization |
| BREEDS | Random horizontal flip, random resized crop, randaugment |
| Amazon | DistilBERT Tokenizer |
| SkinLesionPixels | Normalization |
| UTKFace | Normalization |

## B.3 Evaluation Metrics

Next, we will describe the details of our OOD score calculation. For image datasets, we pass the image inputs through a pretrained ResNet18 ImageNet featurizer to get feature representations, and train a linear classifier to classify whether the feature representations are from the training distribution or the evaluation distribution. We balance the training data of the classifier such that each distribution makes up 50 percent. We then evaluate the linear classifier on the evaluation distribution, and calculate the average predicted likelihood that the batch of inputs are sampled from the evaluation distribution, which we use as the OOD score. For text datasets, we use a similar approach, but use a DistilBERT classifier and Tokenizer instead of the linear classifier and ImageNet featurizer.

There are some limitations to the OOD score. Ideally, we would like the OOD score to be a measure of how well model trained on the training distribution will generalize to the evaluation distribution. This is often the case, such as the datasets in our experiments, but not always. Consider a scenario where the evaluation dataset is a subset of the training dataset with a particular type of feature. Here, a neural network model trained on the training dataset will likely generalize well to the evaluation dataset in terms of task performance. However, the evaluation dataset will likely receive a high OOD score, because the evaluation inputs will be distinguishable from the training inputs, since the evaluation dataset has a high concentration of a particular type of feature. In this case, the OOD score is not a good measure of the distribution shift of the evaluation dataset.

Additionally, with regards to our measure of distance between model predictions and the OCS, we note that this measure is only informative if the different datasets being evaluated have around the same distribution of labels. This is because both the MSE and the KL metrics are being averaged over the evaluation dataset.

## B.4 Characterizing the OCS for Common Loss Functions

In this section, we will precisely characterize the OCS for a few of the most common loss functions: cross entropy, mean squared error (MSE), and Gaussian negative log likelihoog (Gaussian NLL).

**Cross entropy.** With a cross entropy loss, the neural network outputs a vector where each entry is associated with a class, which we denote as $f_\theta(x)_i$. This vector parameterizes a categorical distribution: $P_\theta(y_i|x) = \frac{e^{f_\theta(x)_i}}{\sum_{1 \le j \le m} e^{f_\theta(x)_j}}$. The loss function minimizes the divergence between $P_\theta(y|x)$ and $P(y|x)$, given by

$$\mathcal{L}(f_\theta(x), y) = \sum_{i=1}^{m} \mathbb{1}[y = y_i] \log\Big( \frac{e^{f_\theta(x)_i}}{\sum_{j=1}^{m} e^{f_\theta(x)_j}} \Big).$$

While there can exist multiple optimal constant solutions for the cross entropy loss, they all map to the same distribution which matches the marginal empirical distribution of the training labels, $P_{f^*_{\text{constant}}}(y_i) = \frac{e^{f^*_{\text{constant},i}}}{\sum_{1 \le j \le m} e^{f^*_{\text{constant},j}}} = \frac{1}{N} \sum_{1 \le i \le N} \mathbb{1}[y = y_i]$.

For the cross entropy loss, the uncertainty of the neural network prediction can be captured by the entropy of the output distribution. Because $P_{f^*_{\text{constant}}}(y)$ usually has much higher entropy than $P_\theta(y|x)$ evaluated on the training distribution, $P_\theta(y|x)$ on OOD inputs will tend to have higher entropy than on in-distribution inputs.

**Gaussian NLL.** With this loss function, the output of the neural network parameterizes the mean and standard deviation of a Gaussian distribution, which we denote as $f_\theta(x) = [\mu_\theta(x), \sigma_\theta(x)]$. The objective is to minimize the negative log likelihood of the training labels under the predicted distribution, $P_\theta(y|x) \sim \mathcal{N}(\mu_\theta(x), \sigma_\theta(x))$:

$$\mathcal{L}(f_\theta(x), y) = \log(\sigma_\theta(x)^2) + \frac{(y - \mu_\theta(x))^2}{\sigma_\theta(x)^2}.$$

Let us similarly denote $f^*_{\text{constant}} = [\mu^*_{\text{constant}}, \sigma^*_{\text{constant}}]$. In this case, $\mu^*_{\text{constant}} = \frac{1}{N} \sum_{1 \le i \le N} y_i$, and $\sigma^*_{\text{constant}} = \frac{1}{N} \sum_{1 \le i \le N} (y_i - \mu^*_{\text{constant}})^2$. Here, $\sigma^*_{\text{constant}}$ is usually much higher than the standard deviation of $P(y|x)$ for any given $x$. Thus, our observation suggests that neural networks should predict higher standard deviations for OOD inputs than training inputs.

**MSE.** Mean squared error can be seen as a special case of Gaussian NLL in which the network only predicts the mean of the Gaussian distribution, while the standard deviation is held constant, i.e. $P_\theta(y|x) \sim \mathcal{N}(f_\theta(x), 1)$. The specific loss function is given by:

$$\mathcal{L}(f_\theta(x), y) = (y - f_\theta(x))^2.$$

Here, $f^*_{\text{constant}} = \frac{1}{N} \sum_{1 \le i \le N} y_i$. Unlike the previous two examples, in which OOD predictions exhibited greater uncertainty, predictions from an MSE loss do not capture any explicit notions of uncertainty. However, our observation suggests that the model's predicted mean will still move closer to the average label value as the test-time inputs become more OOD.

## C  EMPIRICAL ANALYSIS

### C.1  ADDITIONAL EXPERIMENTS

In this section, we will provide additional experimental analysis to support the hypothesis that we put forth in Section 4. First, in order to understand whether the trends that we observe for CIFAR10 and MNIST would scale to larger models and datasets, we perform the same analysis as the ones presented in Figure 4 on a ResNet50 model trained on ImageNet, and evaluated on ImageNet-Sketch and ImageNet-R(endition). Our findings are presented in Figure 10. Here, we can see that the same trends from the CIFAR10 and MNIST analysis seem to transfer to the ImageNet models.

Next, we aim to better understand the effects of normalization layers in the network on the behavior of the model representations. We trained models with no normalization (NN), batch normalization (BN), and layer normalization (LN) on the MNIST and CIFAR10 datasets, and evaluated them on OOD test sets with increasing levels of rotation and noise. Note that the model architecture and all

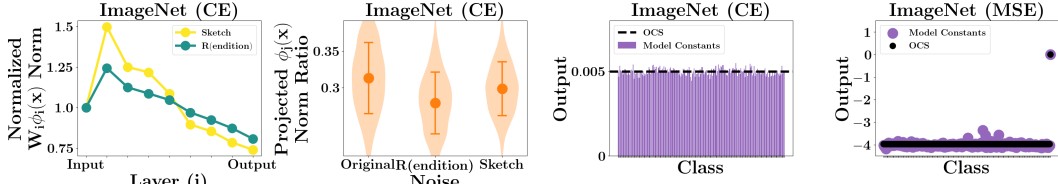

Figure 10: Analysis of the interaction between representations and weights for as distribution shift increases, for a model trained on ImageNet and evaluated on ImageNet-Sketch and ImageNet-R(enditions).

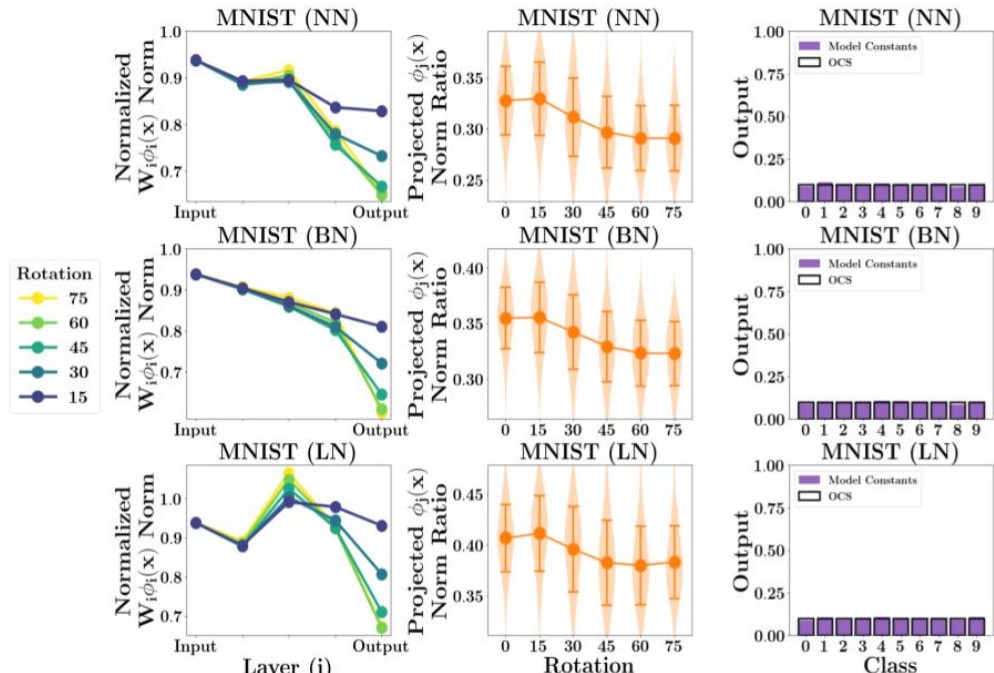

Figure 11: Analysis of the interaction between representations and weights as distribution shift increases, for a model trained on MNIST and evaluated on increasing levels of rotation. The models being considered includes a four layer neural network with no normalization (top), batch normalization (middle), and layer normalization (bottom).

other training details are held fixed across these models (for each datasets) with the exception of the type of normalization layer used (or lack thereof). We perform the same analysis as the ones presented in Figure 4, and present our findings in Figure 11 and 12. We found similar trends across the different models which are consistent with the ones we presented in Figure 4

## C.2 ANALYSIS DETAILS

In this section, we will provide details on the specific neural network layers that we used in our analysis in Sections 4.1 and C.1. We will illustrate diagrams for the neural network architectures that we used for each of our experiments, along with labels of the layers associated with each quantity we measure. In the first column of each analysis figure, we measure quantities at different layers of of the network; we denote these by $i_0, ..., i_n$, where each $i$ represents one tick in the X-axis from left to right. We use $j$ to denote the layer used in the plots in second column of each figure. We $k_{CE}$ (and $k_{MSE}$)to denote the layer used in the plots in the third (and fourth) columns of each figure, respectively. We illustrate the networks used in Figure 4 in Figure 13, the one used in Figure 10 in Figure 14, the ones used in Figure 11 in Figure 15, and the ones used in Figure 12 in Figure 16.

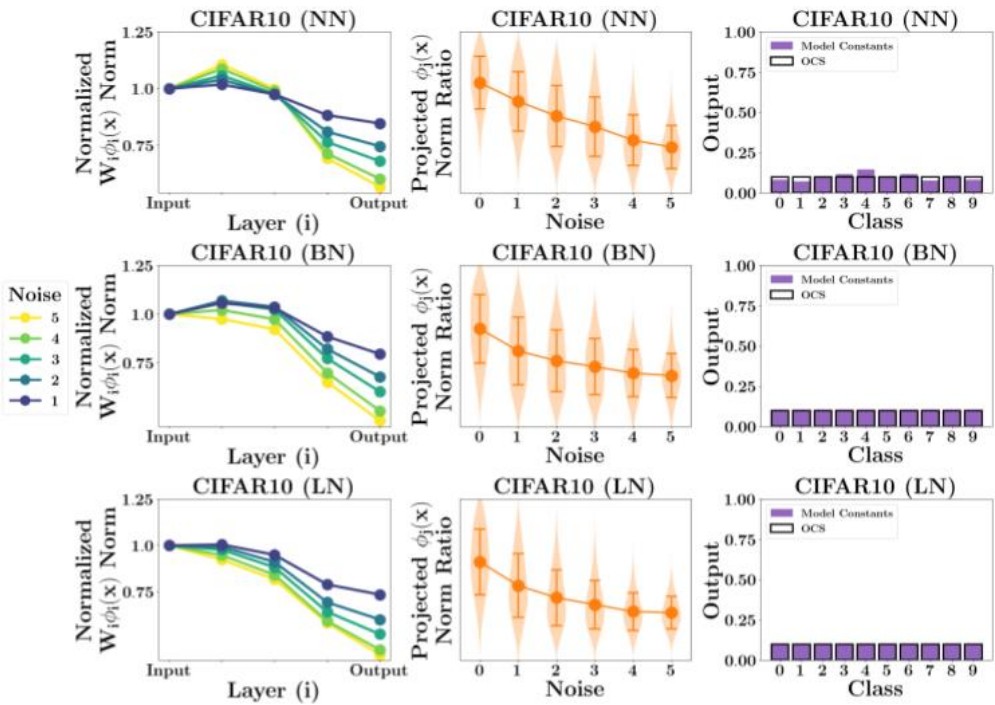

Figure 12: Analysis of the interaction between representations and weights as distribution shift increases, for a model trained on CIFAR10 and evaluated on increasing levels of noise. The models being considered includes AlexNet with no normalization (top), batch normalization (middle), and layer normalization (bottom).

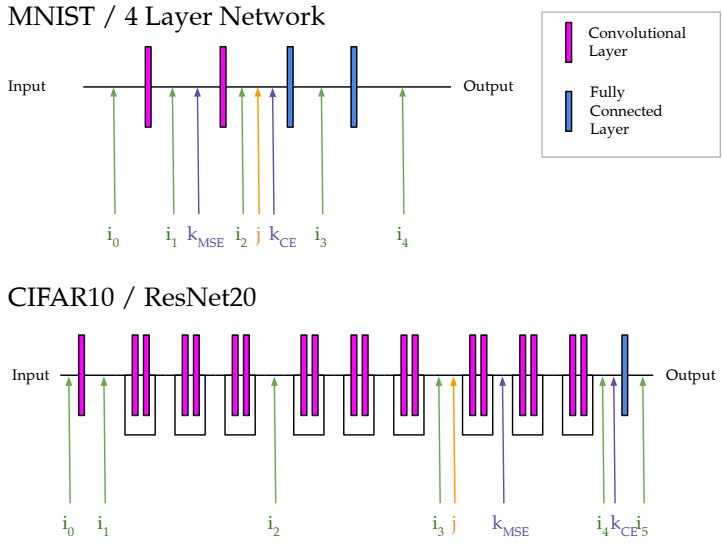

Figure 13: Diagram of neural network models used in our experimental analysis in Figure 4, along with labels of the specific layers we used in our analysis.

## D PROOFS FROM SECTION 4.2

In the first subsection, we describe the setup, and gradient flow with some results from prior works that we rely upon in proving our claims on the in-distribution and out-of-distribution activation magnitudes. In the following subsections we prove our main claims from Section 4.2.

ImageNet / ResNet50

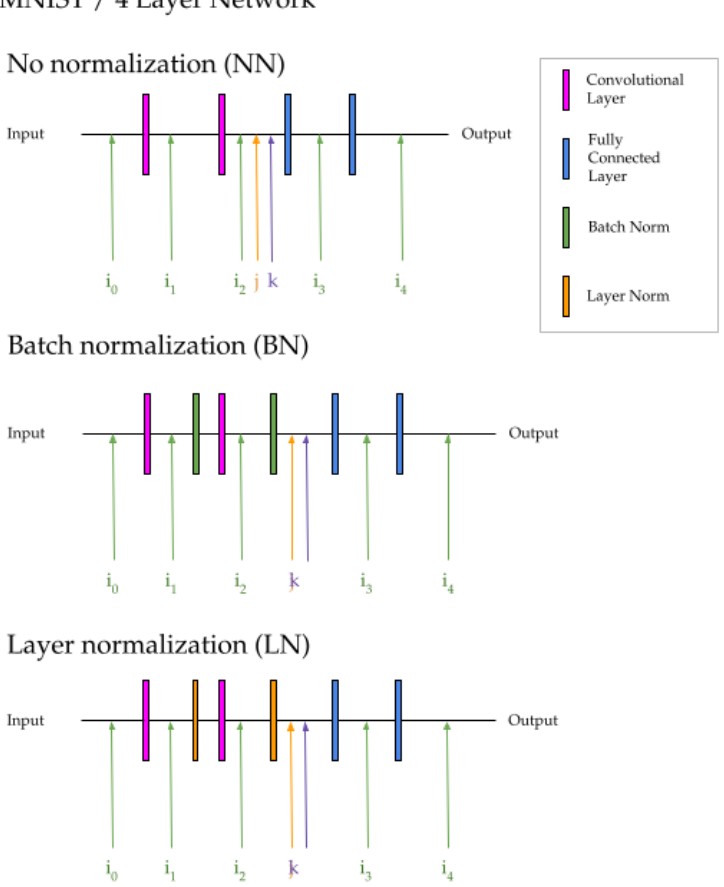

Figure 14: Diagram of neural network models used in our experimental analysis in Figure 10, along with labels of the specific layers we used in our analysis.

MNIST / 4 Layer Network

Figure 15: Diagram of neural network models used in our experimental analysis in Figure 11, along with labels of the specific layers we used in our analysis.

**Setup.** We are learning over a class of homogeneous neural networks $\mathcal{F} := \{f(W; x) : w \in \mathcal{W}\}$, with $L$ layers, and element wise activation $\sigma(x) = x \mathbb{1}(x \geq 0)$ (ReLU function), taking the functional form:

$$f(W; x) = W_L \sigma(W_{L-1} \ldots \sigma(W_2 \sigma(W_1 x)) \ldots),$$

where $W_i \in \mathbb{R}^{m \times m}, \forall i \in \{2, \ldots, L-1\}$, $W_1 \in \mathbb{R}^{m \times 1}$ and output dimension is set to 1, i.e., $W_L \in \mathbb{R}^{1 \times m}$. We say that class $\mathcal{F}$ is homogeneous, if there exists a constant $C$ such that, for all

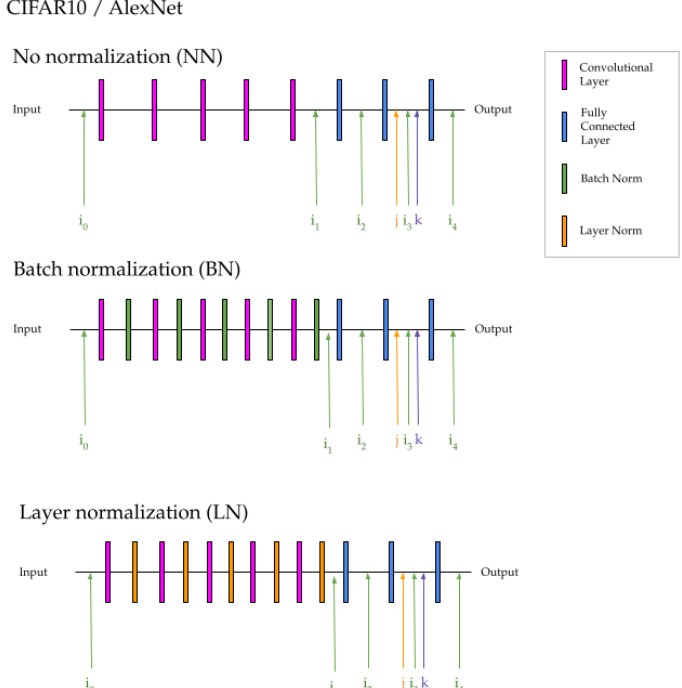

Figure 16: Diagram of neural network models used in our experimental analysis in Figure 12, along with labels of the specific layers we used in our analysis.

$w \in \mathcal{W}$, we have:

$$f(\alpha \cdot W; x) = \alpha^C \cdot f(W; x).$$

Our focus is on a binary classification problem where we have a joint distribution over inputs and labels: $\mathcal{X} \times \mathcal{Y}$. Here, the inputs are from set $\mathcal{X} := \{x \in \mathbb{R}^d : \|x\|_2 \leq B\}$, and labels are binary $\mathcal{Y} := \{-1, +1\}$. We have an IID sampled training dataset $\mathcal{D} := \{(x_i, y_i)\}_{i=1}^n$ containing pairs of data points $x_i$ and their corresponding labels $y_i$.

For a loss function $\ell : \mathbb{R} \to \mathbb{R}$, the empirical loss of $f(W; x)$ on the dataset $\mathcal{D}$ is

$$L(w; \mathcal{D}) := \sum_{i=1}^n \ell\left(y_i f(W; x_i)\right). \tag{1}$$

Here, the loss $\ell$ can be the exponential loss $\ell(q) = e^{-q}$ and the logistic loss $\ell(q) = \log(1 + e^{-q})$. To refer to the output (post activations if applicable) of layer $j$ in the network $f(W; \cdot)$, for the input $x$, we use the notation: $f_j(W; x)$.

**Gradient flow (GF).** We optimize the objective in Equation equation 1 using gradient flow. Gradient flow captures the behavior of gradient descent with an infinitesimally small step size (Arora et al., 2019; Huh et al., 2021; Galanti et al., 2022; Timor et al., 2023). Let $W(t)$ be the trajectory of gradient flow, where we start from some initial point $W(0)$ of the weights, and the dynamics of $W(t)$ is given by the differential equation:

$$\frac{dW(t)}{dt} = -\nabla_{W=W(t)} L(w; \mathcal{D}). \tag{2}$$

Note that the ReLU function is not differentiable at 0. Practical implementations of gradient methods define the derivative $\sigma'(0)$ to be some constant in $[0, 1]$. Following prior works (Timor et al., 2023), in this work we assume for convenience that $\sigma'(0) = 0$. We say that gradient flow converges if the following limit exists:

$$\lim_{t \to \infty} W(t).$$

In this case, we denote $W(\infty) := \lim_{t \to \infty} W(t)$. We say that the gradient flow converges in direction if the following limit exists:

$$\lim_{t \to \infty} W(t)/\|W(t)\|_2.$$

Whenever the limit point $\lim_{t \to \infty} W(t)$ exists, we refer to the limit point as the ERM solution by running gradient flow, and denote it as $\hat{W}$.

**Gradient flow convergence for interpolating homogeneous networks.** Now, we use a result from prior works that states the implicit bias of gradient flow towards max-margin solutions when sufficiently deep and wide homogeneous networks are trained with small learning rates and exponential tail classification losses. As the loss converges to zero, the solution approaches a KKT point of an optimization problem that finds the minimum $l_2$ norm neural network with a margin of at least 1 on each point in the training set. This is formally presented in the following Lemma adapted from (Ji & Telgarsky, 2020) and (Lyu & Li, 2019).

**Lemma D.1 (Gradient flow is implicitly biased towards minimum $\|\cdot\|_2$)** *Consider minimizing the average of either the exponential or the logistic loss (in equation 1) over a binary classification dataset $\mathcal{D}$ using gradient flow (in equation 2) over the class of homogeneous neural networks $\mathcal{F}$ with ReLU activations. If the average loss on $\mathcal{D}$ converges to zero as $t \to \infty$, then gradient flow converges in direction to a first order stationary point (KKT point) of the following maximum margin problem in the parameter space of $\mathcal{W}$:*

$$\min_{f(W;x) \in \mathcal{F}} \frac{1}{2} \sum_{j \in [L]} \|W_i\|_2^2 \quad s.t. \quad \forall i \in [n] \ \ y_i f(W; x_i) \geq 1. \tag{3}$$

**Spectrally normalized margin based generalization bounds (Bartlett et al., 2017).** Prior work on Rademacher complexity based generalization bounds provides excess risk bounds based on spectrally normalized margins, which scale with the Lipschitz constant (product of spectral norms of weight matrices) divided by the margin.

**Lemma D.2 (Adaptation of Theorem 1.1 from (Bartlett et al., 2017))** *For the class homogeneous of ReLU networks in $\mathcal{F}$ with reference matrices $(A_1, \ldots, A_L)$, and all distributions $P$ inducing binary classification problems, i.e., distributions over $\mathbb{R}^d \times \{-1, +1\}$, with probability $1 - \delta$ over the IID sampled dataset $\mathcal{D}$, and margin $\gamma > 0$, the network $f(w; x)$ has expected margin loss upper bounded as:*

$$\mathbb{E}_{(x,y) \sim P} \mathbb{1}(yf(w; x) \geq \gamma) \ \leq \ \frac{1}{n} \sum_{i=1}^{n} \mathbb{1}(y_i f(w; x_i) \geq \gamma) + \tilde{\mathcal{O}}\left(\frac{\mathcal{R}_{W,A}}{\gamma n} \log(m) + \sqrt{\frac{\log(1/\delta)}{n}}\right),$$

*where the covering number bound determines $\mathcal{R}_{W,A} := \left(\prod_{i=1}^{L} \|W_i\|_{\mathrm{op}}\right)\left(\sum_{i=1}^{L} \frac{\|W_i^\top - A_i^\top\|_{2,1}}{\|W_i\|_{\mathrm{op}}^{2/3}}\right)$.*

## D.1 LOWER BOUND FOR ACTIVATION MAGNITUDE ON IN-DISTRIBUTION DATA

**Proposition D.1 ($P_{\mathrm{train}}$ observes high norm features)** *When $f(\hat{W}; x)$ fits $\mathcal{D}$, i.e., $y_i f(\hat{W}; x_i) \geq \gamma$, $\forall i \in [N]$, then w.h.p $1 - \delta$ over $D$, layer $j$ representations $f_j(\hat{W}; x)$ satisfy $\mathbb{E}_{P_{\mathrm{train}}}[\|f_j(\hat{W}; x)\|_2] \geq (1/C_0)(\gamma - \tilde{\mathcal{O}}(\sqrt{\log(1/\delta)/N} + C_1^2 \log m/N\gamma))$, if $\exists$ constants $C_0, C_1$ s.t. $\|\hat{W}_j\|_2 \leq C_0^{j+1-L}$, $C_1 \geq C_0^L$.*

*Proof.*

Here, we lower bound the expected magnitude of the in-distribution activation norms at a fixed layer $j$ in terms of the expected activation norm at the last layer, by repeatedly applying Cauchy-Schwartz

inequality and the property of ReLU activations: $\|\sigma(x)\|_2 \le \|x\|_2$, alternatively.

$$
\begin{aligned}
&\mathbb{E}_{P_{\text{train}}} |f(\hat{W}; x)| \\
&= \quad \mathbb{E}_{P_{\text{train}}} \left[ \|\hat{W}_L \sigma(\hat{W}_{L-1} \dots \sigma(\hat{W}_2 \sigma(\hat{W}_1 x)) \dots)\|_2 \right] \\
&\le \quad \|\hat{W}_L\|_{\text{op}} \mathbb{E}_{P_{\text{train}}} \left[ \|\sigma(\hat{W}_{L-1} \dots \sigma(\hat{W}_2 \sigma(\hat{W}_1 x)) \dots)\|_2 \right] \quad \text{(Cauchy-Schwartz)} \\
&\le \quad \|\hat{W}_L\|_{\text{op}} \mathbb{E}_{P_{\text{train}}} \left[ \|\hat{W}_{L-1} \dots \sigma(\hat{W}_2 \sigma(\hat{W}_1 x))\|_2 \right] \quad \text{(ReLU activation property)}
\end{aligned}
$$

Doing the above repeatedly gives us the following bound:

$$
\begin{aligned}
\mathbb{E}_{P_{\text{train}}} |f(\hat{w}; x)| &\le \left( \prod_{k=j+1}^{L} \|\hat{W}_k\|_{\text{op}} \right) \cdot \mathbb{E}_{P_{\text{train}}} \left[ \|f_j(\hat{W}; x)\|_2 \right] \\
&\le C_0^{L-(j+1)/L} \left[ \|f_j(\hat{W}; x)\|_2 \right] \le C_0 \left[ \|f_j(\hat{W}; x)\|_2 \right]
\end{aligned}
$$

Next, we use a generalization bound on the margin loss to further lower bound the expected norm of the last layer activations. Recall, that we use gradient flow to converge to globally optimal solution of the objective in equation 1 such that the training loss converges to 0 as $t \to \infty$. Now, we use the spectrally normalized generalization bound from Lemma D.2 to get:

$$
\mathbb{E}_{(x,y) \sim P} \mathbb{1}(y f(\hat{W}; x) \ge \gamma) \lesssim \tilde{\mathcal{O}} \left( \frac{\log m}{\gamma N} \left( \prod_{i=1}^{L} \|\hat{W}_i\|_{\text{op}} \right) \left( \sum_{i=1}^{L} \frac{\|\hat{W}_i^\top - A_i^\top\|_{2,1}}{\|\hat{W}_i\|_{\text{op}}^{2/3}} \right) + \sqrt{\frac{\log(1/\delta)}{N}} \right)
$$

This implies that with probability $1 - \delta$ over the training set $\mathcal{D}$, on at least $\mathcal{O}(\sqrt{\log(1/\delta)/n})$ fraction of the test set, the margin is at least $\gamma$, i.e., if we characterize the set of correctly classified test points as $\mathcal{C}_{\hat{W}}$, then:

$$
\begin{aligned}
\mathbb{E}[|f(\hat{W}; x)| \mid (x,y) \in \mathcal{C}_{\hat{W}}] &= \mathbb{E}[|y \cdot f(\hat{W}; x)| \mid (x,y) \in \mathcal{C}_{\hat{W}}] \\
&\ge \mathbb{E}[y \cdot f(\hat{W}; x) \mid (x,y) \in \mathcal{C}_{\hat{W}}] \ge \gamma
\end{aligned}
$$

We are left with lower bounding: $\mathbb{E}[|f(\hat{W}; x)| \mid (x,y) \notin \mathcal{C}_{\hat{W}}]$ which is trivially $\ge 0$. Now, from the generalization guarantee we know:

$$
\begin{aligned}
\mathbb{E}(\mathbb{1}((x,y) \in C_{\hat{W}})) &\lesssim \tilde{\mathcal{O}} \left( \frac{\log m}{\gamma N} \left( \prod_{i=1}^{L} C_0^{1/L} \right) \left( \sum_{i=1}^{L} \frac{\|\hat{W}_i^\top - A_i^\top\|_{2,1}}{\|\hat{W}_i\|_{\text{op}}^{2/3}} \right) + \sqrt{\frac{\log(1/\delta)}{N}} \right) \\
&\lesssim \tilde{\mathcal{O}} \left( \frac{\log m}{\gamma N} C_0 \left( \sum_{i=1}^{L} \frac{\|\hat{W}_i^\top - A_i^\top\|_{2,1}}{\|\hat{W}_i\|_{\text{op}}^{2/3}} \right) + \sqrt{\frac{\log(1/\delta)}{N}} \right) \\
&\lesssim \tilde{\mathcal{O}} \left( \frac{\log m}{\gamma N} C_1 \left( \sum_{i=1}^{L} \frac{\|\hat{W}_i^\top - A_i^\top\|_{2,1}}{\|\hat{W}_i\|_{\text{op}}^{2/3}} \right) + \sqrt{\frac{\log(1/\delta)}{N}} \right),
\end{aligned}
$$

where the final inequality uses

$$
\frac{1}{L} \sum_i \|W_i\|_2^{2/3} \ge \left( \prod_{i=1}^{L} \|W_i\|_2^{2/3} \right)^{1/L},
$$

which is the typical AM-GM inequality. We also use the inequality: $C_1 \ge C_0^{3L/2}$. This bound tells us that:

$$
\mathbb{E}(\mathbb{1}((x,y) \in C_{\hat{W}})) \ge 1 - \tilde{\mathcal{O}} \left( \frac{\log m}{\gamma N} C_1 + \sqrt{\frac{\log(1/\delta)}{N}} \right).
$$

Plugging the above into the lower bound we derived completes the proof of Proposition D.1.

### D.2 Upper bound for activation magnitude on out-of-distribution data

**Theorem D.3 (Feature norms can drop easily on $P_{\text{OOD}}$)** *If $\exists$ a shallow network $f'(W; x)$ with $L'$ layers and $m'$ neurons satisfying conditions in Proposition 4.1 ($\gamma=1$), then optimizing the training objective with gradient flow over a class of deeper and wider homogeneous network $\mathcal{F}$ with $L > L', m > m'$ would converge directionally to a solution $f(\hat{W}; x)$, for which the following is true: $\exists$ a set of rank 1 projection matrices $\{A_i\}_{i=1}^{L}$, such that if representations for any layer $j$ satisfy $\mathbb{E}_{P_{\text{OOD}}}\|A_j f_j(\hat{W}; x)\|_2 \le \epsilon$, then $\exists C_2$ for which $\mathbb{E}_{P_{\text{OOD}}}[|f(\hat{W}; x)|] \lesssim C_0(\epsilon + C_2^{-1/L}\sqrt{L+1/L})$.*

*Proof.*

Here, we show that there will almost rank one subspaces for each layer of the neural network, such that if the OOD representations deviate even slightly from the low rank subspace at any given layer, the last layer magnitude will collapse. This phenomenon is exacerbated in deep and wide networks, since gradient descent on deep homogeneous networks is biased towards KKT points of a minimum norm, max-margin solution (Lyu & Li, 2019), which consequently leads gradient flow on sufficiently deep and wide networks towards weight matrices that are low rank (as low as rank 1).

We can then show by construction that there will always exist these low rank subspaces which the OOD representations must not deviate from for the last layer magnitudes to not drop. Before we prove the main result, we adapt some results from Timor et al. (2023) to show that gradient flow is biased towards low rank solutions in our setting. This is formally presented in Lemma D.4.

**Lemma D.4 (GF on deep and wide nets is learns low rank $W_1, \ldots, W_L$)** *We are given the IID sampled dataset for the binary classification task defined by distribution $P_{\text{train}}$, i.e., $\mathcal{D} := \{(x_i, y_i)\}_{i=1}^{n} \subseteq \mathbb{R}^d \times \{-1, 1\}$. Here, $\|x_i\|_2 \le 1$, with probability 1. Let there exist a homogeneous network in class $\mathcal{F}'$, of depth $L' \ge 2$, and width $m' \ge 2$, if there exists a neural network $f(W'; x)$, such that: $\forall (x_i, y_i) \in \mathcal{D}, f(W'; x_i) \cdot y_i \ge \gamma$, and the weight matrices $W'_1, \ldots, W'_{L'}$ satisfy $\|W'_i\|_F \le C$, for some fixed constant $C > 0$. If the solution $f(W^\star, x)$ of gradient flow any class $\mathcal{F}$ of deeper $L > L'$ and wider $m > m'$ networks $f(W; x)$ converges to the global optimal of the optimization problem:*

$$\min_{f(W;x) \in \mathcal{F}} \frac{1}{2} \sum_{j \in [L]} \|W_i\|_2^2 \quad s.t. \quad \forall i \in [n] \ y_i f(W; x_i) \ge 1, \tag{4}$$

*then for some universal constant $C_1$, the following is satisfied:*

$$\max_{i \in [L]} \|W_i^\star\|_{\text{op}}/\|W_i^\star\|_F \ \ge \ \frac{1}{L} \sum_{i=1}^{L} \frac{\|W_i^\star\|_{\text{op}}}{\|W_i^\star\|_F} \ \ge \ C_1^{1/L} \cdot \sqrt{\frac{L}{L+1}}. \tag{5}$$

*Proof.*

We will prove this using the result from Lemma D.1, and major parts of the proof technique is a re-derivation of some of the results from Timor et al. (2023), in our setting. From Lemma D.1 we know that gradient flow on $\mathcal{F}$ necessarily converges in direction to a KKT point of the optimization problem in Equation 4. Furthermore, from Lyu & Li (2019) we know that this optimization problem satisfies the Mangasarian-Fromovitz Constraint Qualification (MFCQ) condition, which means that the KKT conditions are first-order necessary conditions for global optimality.

We will first construct a wide and deep network $f(W; x) \in \mathcal{F}$, using the network $f(W'; x)$ from the relatively shallower class $\mathcal{F}'$, and then argue about the Frobenius norm weights of the constructed network to be larger than the global optimal of problem in Equation 4.

Recall that $f(W'; x) \in \mathcal{F}'$ satisfies the following:

$$\forall (x_i, y_i) \in \mathcal{D}, \ f(W'; x_i) \cdot y_i \ \ge \ \gamma.$$

Now, we can begin the construction of $f(W; x) \in \mathcal{F}$. Set the scaling factor:

$$\alpha = \left(\sqrt{2}/C\right)^{\frac{L-L'}{L}}.$$

Then, for any weight matrix $W_i$ for $i \in 1, \ldots, L' - 1$, set the value for $W_i$ to be:

$$W_i = \alpha \cdot W_i' = \left(\sqrt{2}/C\right)^{\frac{L-L'}{L}} \cdot W_i'.$$

Let $v$ be the vector of the output layer $L'$ in shallow $f(W'; x)$. Note that this is an $m$-dimensional vector, since this is the final layer for $f(W'; x)$. But in our construction, layer $L'$ is a layer that includes a fully connected matrix $W_{L'} \in \mathbb{R}^{m \times m}$ matrix.

So for layer $L'$, we set the new matrix to be:

$$W_{L'} = \alpha \cdot \begin{bmatrix} v^\top \\ -v^\top \\ \mathbf{0}_m \\ \vdots \\ \mathbf{0}_m \end{bmatrix},$$

where $\mathbf{0}_m$ is the $m$-dimensional vector of 0s.

This means that for the $L'$-th layer in $f(W; x)$ we have the following first two neurons: the neuron that has weights which match the corresponding layer from $f(W'x)$, and the neuron that has weights given by its negation. Note that since the weights in $f(W; x)$ are constructed directly from the weights of $f(W'; x)$, via scaling the weights through the scaling parameter $\alpha$ defined above, we can satisfy the following for every input $x$ for the output of layer $L'$ in $f(W; x)$:

$$f_{L'}(W; x) = \begin{bmatrix} \alpha^k \cdot f(W; x) \\ -\alpha^k \cdot f(W; x) \\ \mathbf{0}_m \\ \vdots \\ \mathbf{0}_m \end{bmatrix}.$$

Next, we define the weight matrices for the layers: $\{L' + 1, \ldots, L\}$. We set the weight matrices $i \in \{L' + 1, \ldots, L - 1\}$ to be:

$$W_i = \left(\frac{\sqrt{2}}{C}\right)^{-\frac{L'}{L}} \cdot \mathbf{I}_m,$$

where $\mathbf{I}_m$ is the $m \times m$ identity matrix. The last layer $L$ in $f(W; x)$ is set to be $\left[\left(\frac{\sqrt{2}}{C}\right)^{-\frac{L'}{L}}, -\left(\frac{\sqrt{2}}{C}\right)^{-\frac{L'}{L}}, 0, \ldots, 0\right]^\top \in \mathbb{R}^m$. For this construction, we shall now prove that $f(W'; x) = f(W; x)$ for every input $x$.

For any input $x$, the output of layer $L'$ in $f(W'x)$ is:

$$\begin{pmatrix} \text{ReLU}\left(\alpha^k \cdot f(W; x)\right) \\ \text{ReLU}\left(-\alpha^k \cdot f(W; x)\right) \\ \mathbf{0}_m \\ \vdots \\ \mathbf{0}_m \end{pmatrix}.$$

Given our construction for the layers that follow we get for the last but one layer:

$$\begin{pmatrix} \left(\left(\frac{\sqrt{2}}{C}\right)^{-\frac{L'}{L}}\right)^{L-L'-1} \cdot \text{ReLU}\left(\alpha^k \cdot f(W; x)\right) \\ \left(\left(\frac{\sqrt{2}}{C}\right)^{-\frac{L'}{L}}\right)^{L-L'-1} \cdot \text{ReLU}\left(-\alpha^k \cdot f(W; x)\right) \\ \mathbf{0}_m \\ \vdots \\ \mathbf{0}_d \end{pmatrix}.$$

Hitting the above with the last layer $[\left(\frac{\sqrt{2}}{C}\right)^{-\frac{L'}{L}}, -\left(\frac{\sqrt{2}}{C}\right)^{-\frac{L'}{L}}, 0, \ldots, 0]^\top$, we get:

$$\left(\left(\frac{\sqrt{2}}{C}\right)^{-\frac{L'}{L}}\right)^{L-L'-1} \cdot \mathrm{ReLU}\left(\alpha^k \cdot f(W;x)\right)$$

$$-\left(\left(\frac{\sqrt{2}}{C}\right)^{-\frac{L'}{L}}\right)^{L-L'-1} \cdot \mathrm{ReLU}\left(-\alpha^k \cdot f(W;x)\right)$$

$$=\left(\frac{\sqrt{2}}{C}\right)^{-\frac{L'}{L}\cdot(L-L')} \cdot \left(\frac{\sqrt{2}}{C}\right)^{\frac{L-L'}{L}\cdot L'} \cdot f(W;x) = f(W;x).$$

Thus, $f(W;x) = f(W';x)$, $\forall x$.

If $W = [W_1, \ldots, W_L]$ be the parameters of wider and deeper network $f(W;x)$ and if $fW^\star; x$ be the network with the parameters $W^\star$ achieving the global optimum of the constrained optimization problem in equation 4.

Because we have that $f(W^\star;x)$ is of depth $L > L'$ and has $m$ neurons where $m > m'$ and is optimum for the squared $\ell_2$ norm minimization problem, we can conclude that $\|W^\star\|_2 \leq \|W\|$. Therefore,

$$\|W^\star\|^2 \leq \|W'\|^2$$

$$=\left(\sum_{i=1}^{L'-1}\left(\left(\sqrt{2}/c\right)^{L-L'/L\cdot L'}\right)^2 \|W_i\|_F^2\right)$$

$$+\left(\left(\sqrt{2}/c\right)^{L-L'/L\cdot L'}\right)^2\left(2\|W_k\|_F^2\right)$$

$$+(L-L'-1)\left(\left(\sqrt{2}/c\right)^{L'-L/L\cdot L'}\right)^2\cdot 2+\left(\left(\sqrt{2}/c\right)^{L'-L/L\cdot L'}\right)^2\cdot 2$$

$$\leq\left(\left(\sqrt{2}/c\right)^{L-L'/L\cdot L'}\right)^2 C^2(L'-1)+\left(\left(\sqrt{2}/c\right)^{L-L'/L\cdot L'}\right)^2\cdot 2C^2$$

$$+(2(L-L'-1)+2)\left(\left(\sqrt{2}/c\right)^{L'-L/L\cdot L'}\right)^2$$

$$=C^2(L'+1)\left(\left(\sqrt{2}/c\right)^{L-L'/L\cdot L'}\right)^2\left(\left(\sqrt{2}/c\right)^{L-L'/L\cdot L'}\right)^2$$

$$+2(L-L')\left(\left(\sqrt{2}/c\right)^{L'-L/L\cdot L'}\right)^2$$

$$=(2/c^2)^{L-L'/L} C^2(L'+1)+(2/c^2)^{-L'/L}\cdot 2(L-L')$$

$$=2\cdot(2/B^2)^{-L'/L}(L'+1)+(2/c^2)^{-L'/L}\cdot 2(L-L')=2\cdot(2/c^2)^{-L'/L}(L+1).$$

Since $f^\star$ is a global optimum of equation 4, we can also show that it satisfies:

$$\|W_i^\star\|_F = \|W_j^\star\|_F, \quad i < j, \quad i,j \in [L]$$

By the lemma, there is $C^\star > 0$ such that $C^\star = \|W_i^\star\|_F$ for all $i \in [L]$. To see why this is true, consider the net $f(\tilde{W};x)$ where $\tilde{W}_i = \eta W_i^\star$ and $\tilde{W}_j = 1/\eta W_j^\star$, for some $i < j$ and $i,j \in [L]$. By the property of homogeneous networks we have that for every input $x$, we get: $f(\tilde{W};x) = f(W^\star;x)$. We can see how the sum of the weight norm squares change with a small change in $\eta$:

$$\frac{d}{d\eta}(\eta^2\|W_i^\star\|+(1/\eta^2)\|W_j^\star\|_2^2)=0$$

at $\eta = 1$, since $W^\star$ is the optimal solution. Taking the derivative we get: $\frac{d}{d\eta}(\eta^2\|W_i^\star\| + (1/\eta^2)\|W_j^\star\|_2^2) = 2\eta\|W_i^\star\|_F^2 - 2/\eta^3\|W_j^\star\|_F^2$. For this expression to be zero, we must have $W_i^\star = W_j^\star$, for any $i < j$ and $i, j \in [L]$.

Based on the above result we can go back to our derivation of $\|W^\star\|_F^2 \leq 2\frac{2}{C^2}^{-L'/L}(L+1)$. Next, we can see that for every $i \in [L]$ we have:

$$C^{\star 2}L \leq 2\frac{2}{C^2}^{-L'/L}(L+1)$$

$$C^{\star 2} \leq 2\frac{2}{C^2}^{-L'/L}\frac{(L+1)}{L}$$

$$1/C^\star \geq \frac{1}{\sqrt{2}}\left(\frac{C}{\sqrt{2}}\right)^{L'/L}\sqrt{L/L+1}$$

Now we use the fact that $\forall x \in \mathcal{X}$, the norm $\|x\|_2 \leq 1$:

$$1 \leq y_i f^\star(x_i) \leq |f^\star(x_i)| \leq \|x_i\|\prod_{i\in[L]}\|W_i^\star\|_{\mathrm{op}} \leq \prod_{i\in[L]}\|W_i^\star\|_{\mathrm{op}} \leq \left(\frac{1}{L}\sum_{i\in[L]}\|W_i^\star\|_{\mathrm{op}}\right)^L,$$

Thus: $\frac{1}{L}\sum_{i\in[L]}\|W_i^\star\|_{\mathrm{op}} \geq 1$. Plugging this into the lower bound on $\frac{1}{C^\star}$:

$$\frac{1}{L}\sum_{i\in[L]}\frac{\|W_i^\star\|_{\mathrm{op}}}{\|W_i^\star\|_F} = \frac{1}{C^\star}\cdot\frac{1}{L}\sum_{i\in[L]}\|W_i^\star\|_{\mathrm{op}} \geq \frac{1}{\sqrt{2}}\left(\frac{C}{\sqrt{2}}\right)^{L'/L}\sqrt{L/L+1}\cdot 1 \tag{6}$$

$$= \frac{1}{\sqrt{2}}\cdot\left(\frac{\sqrt{2}}{C}\right)^{\frac{L'}{L}}\cdot\sqrt{\frac{L}{L+1}}. \tag{7}$$

This further implies that $\forall i \in [L]$:

$$\|W_i^\star\|_F \leq \|W_i^\star\|_{\mathrm{op}}\sqrt{2}\cdot\frac{\sqrt{2}}{C}^{-L'/L}\cdot\sqrt{L+1/L}.$$

Setting $C' = \frac{1}{\sqrt{2}}\cdot\left(\frac{\sqrt{2}}{C}\right)^{L'}$ we get the final result:

$$\|W_i^\star\|_F \leq \|W_i^\star\|_{\mathrm{op}}\cdot C_1^{1/L}\cdot\sqrt{\frac{L+1}{L}}, \quad \forall i \in [n].$$

From Lemma D.4 we know that at each layer the weight matrices are almost rank 1, i.e., for each layer $j \in [L]$, there exists a vector $v_j$, such that $\|v_j\|_2 = 1$ and $W_j \approx \sigma_j v_j v_j^\top$ for some $\sigma_j > 0$. More formally, we know that for any $L > L'$ and $m \geq m'$ satisfying the conditions in Lemma D.4, for every layer $j$ we have:

$$\|(I - v_j v_j^\top)\hat{W}_j\|_2 = \sqrt{\|\hat{W}_j\|_F^2 - \sigma_j^2} \tag{8}$$

Next, we can substitute the previous bound that we derived: $\|\hat{W}_j\|_F \leq \sigma_j C'^{1/L}\cdot\sqrt{\frac{L+1}{L}}$ to get the following, when gradient flow converges to the globally optimal solution of equation 4:

$$\|(I - v_j v_j^\top)\hat{W}_j\|_2 \leq \sigma_j\sqrt{C'^{2/L}\cdot L+1/L - 1} \leq \sigma_j C'^{1/L}\sqrt{\frac{L+1}{L}} \tag{9}$$

Now we are ready to derive the final bound on the last layer activations:

$$\mathbb{E}_{x \sim P_{\text{OOD}}}|f(\hat{W}; x)|_2 = \mathbb{E}_{P_{\text{OOD}}}|W_L \sigma(W_{L-1} \sigma(W_{L-2} \ldots \sigma(W_2 \sigma(W_1 x)) \ldots))|$$

$$= \mathbb{E}_{P_{\text{OOD}}}|W_L \sigma(W_{L-1} \sigma(W_{L-2} \ldots \|f_j(\hat{W})\|_2 \cdot \frac{f_j(\hat{W}x)}{\|f_j(\hat{W}; x)\|} \ldots))|$$

$$= \|f_j(\hat{W})\|_2 \cdot \mathbb{E}_{P_{\text{OOD}}}|W_L \sigma(W_{L-1} \sigma(W_{L-2} \ldots \cdot \frac{f_j(\hat{W}x)}{\|f_j(\hat{W}; x)\|} \ldots))|$$

$$\leq \mathbb{E}_{P_{\text{OOD}}}\|f_j(\hat{W})\|_2 \cdot \prod_{z=j+1}^{L} C_0^{1/L} \leq \mathbb{E}_{P_{\text{OOD}}}\|f_j(\hat{W})\|_2 \cdot C_0$$

where the final inequality repeatedly applies Cauchy-Schwartz on a norm one vector: $f_j(\hat{W}; x)/\|f_j(\hat{W}; x)\|_2$, along with another property of ReLU activations: $\|\sigma(v)\|_2 \leq \|v\|_2$.

$$\mathbb{E}_{P_{\text{OOD}}}\|f_j(\hat{W})\|_2 \leq \sqrt{\mathbb{E}_{P_{\text{OOD}}}\|f_j(\hat{W})\|_2^2}$$

$$\leq \sqrt{\mathbb{E}_{P_{\text{OOD}}}(\sigma_j^2\|v_j v_j^\top f(\hat{W}; x)\|_2^2 + \|(I - v_j v_j^\top)\hat{W}_j\|_2^2 C_0^2)}$$

$$\leq \sqrt{\sigma_j^2 \epsilon^2 + \|(I - v_j v_j^\top)\hat{W}_j\|_2^2 C_0^2}$$

Since, $\mathbb{E}_{P_{\text{OOD}}} \leq \|v_j v_j^\top \hat{W}_j\|_2$ and

$$\mathbb{E}_{P_{\text{OOD}}}\|f_j(\hat{W})\|_2 \leq \sigma_j \sqrt{(\epsilon^2 + \|(I - v_j v_j^\top)f_j(\hat{W})\|_2)}$$

$$\leq \sigma_j \sqrt{(\epsilon^2 + C'^{2/L} \cdot {}^{L+1}/L)}$$

$$\leq \sigma_j(\epsilon + C'^{1/L} \cdot \sqrt{{}^{L+1}/L})$$

Recall that $\sigma_j \leq C_0^{1/L}$. From the above, we get the following result:

$$\mathbb{E}_{x \sim P_{\text{OOD}}}|f(\hat{W}; x)|_2 \leq C_0 \mathbb{E}_{P_{\text{OOD}}}\|f_j(\hat{W})\|_2 \leq C_0(\epsilon + C'^{1/L} \cdot \sqrt{{}^{L+1}/L}),$$

which completes the proof of Theorem D.3.

## D.3  BIAS LEARNT FOR NEARLY HOMOGENEOUS NETS

In this subsection, we analyze a slightly modified form of typical deep homogeneous networks with ReLU activations. To study the accumulation of model constants, we analyze the class of functions $\tilde{\mathcal{F}} = \{f(W; \cdot) + b : b \in \mathbb{R}, f(W; \cdot) \in \mathcal{F}\}$, which consists of deep homogeneous networks with a bias term in the final layer. In Proposition 4.2, we show that there exists a set of margin points (analogous to support vectors in the linear setting) which solely determines the model's bias $\hat{b}$.

**Proposition D.2 (Analyzing network bias)** *If gradient flow on $\tilde{\mathcal{F}}$ converges directionally to $\hat{W}, \hat{b}$, then $\hat{b} \propto \sum_k y_k$ for margin points $\{(x_k, y_k) : y_k \cdot f(\hat{W}; x_k) = \arg\min_{j \in [N]} y_j \cdot f(\hat{W}; x_j)\}$.*

*Proof.*

Lemmas C.8, C.9 from Lyu & Li (2019) can be proven for gradient flow over $\tilde{F}$ as well since all we need to do is construct $h_1, \ldots, h_N$ such that $h_i$ satisfoes forst order stationarity for the $i^{\text{th}}$ constraint in the following optimization problem, that is only lightly modified version of problem instance $P$ in Lyu & Li (2019):

$$\min_{W,b} L(W, b; \mathcal{D}) := \sum_{i=1}^{L} \|W_i\|_F^2$$

$$\text{s.t.} \quad y_i f(W; x_i) \geq 1 - b, \quad \forall i \in [N]$$

Note that the above problem instance also satisfies MFCQ (Mangasarian-Fromovitz Constraint Qualification), which can also be shown directly using Lemma C.7 from Lyu & Li (2019).

As a consequence of the above, we can show that using gradient flow to optimize the objective:

$$\min_{W,b} \frac{1}{N} \sum_{(x,y) \in \mathcal{D}} \exp\left(-y \cdot (f(W;x) + b)\right),$$

also converges in direction to the KKT point of the above optimization problem with the nearly homogeneous networks $\mathcal{F}$. This result is also in line with the result for linear non-homogeneous networks derived in Soudry et al. (2018).

Finally, at directional convergence, the gradient of the loss $\frac{\partial L(W;\mathcal{D})}{\partial W}$ converges, as a direct consequence of the asymptotic analysis in Lyu & Li (2019).

Let us denote the set of margin points at convergence as $\mathcal{M} = \{(x_k, y_k) : y_k f(W; x_k) = \min_{j \in [N]} y_j f(W; x_j)\}$. These, are precisely the set of points for which the constraint in the above optimization problem is tight, and the gradients of their objectives are the only contributors in the construction of $h_1, \ldots, h_N$ for Lemma C.8 in Lyu & Li (2019). Thus, it is easy to see that at convergence the gradient directionally converges to the following value, which is purely determined only by the margin points in $\mathcal{M}$.

$$\lim_{t \to \infty} \frac{\frac{\partial}{\partial W} L(W, b; \mathcal{D})}{\|\frac{\partial}{\partial W} L(W, b; \mathcal{D})\|_2} = -\frac{\sum_{k \in \mathcal{M}} y_k \cdot \nabla_W f(W; x_k)}{\|\sum_{k \in \mathcal{M}} y_k \cdot \nabla_W f(W; x_k)\|_2}$$

Similarly we can take the derivative of the objective with respect to the bias $b$, and verify its direction. For that, we can note that: $\frac{\partial \exp(-y(f(W;x)+b))}{\partial b} = -y \cdot \exp(-y(f(W;x)+b))$, but more importantly, $\exp(-y(f(W;x)+b)$ evaluates to the same value for all margin points in $\mathcal{M}$. Hence,

$$\lim_{t \to \infty} \frac{\frac{\partial}{\partial b} L(W, b; \mathcal{D})}{\|\frac{\partial}{\partial b} L(W, b; \mathcal{D})\|_2} = -\frac{\sum_{k \in \mathcal{M}} y_k}{|\sum_{k \in \mathcal{M}} y_k|}$$

While both $\hat{b}$ and $\hat{W}$ have converged directionally, their norms keep increasing, similar to analysis in other works (Soudry et al., 2018; Huh et al., 2021; Galanti et al., 2022; Lyu & Li, 2019; Timor et al., 2023). Thus, from the above result it is easy to see that bias keeps increasing along the direction that is just given by the sum of the labels of the margin points in the binary classification task. This direction also matches the OCS solution direction (that only depends on the label marginal) if the label marginal distribution matches the distribution of the targets $y$ on the support points. This completes the proof of Proposition D.2.

