# OpenReview forum: "Deep Neural Networks Tend To Extrapolate Predictably"
_ICLR.cc/2024/Conference — ICLR 2024 poster_

### Official Review · Reviewer_5N2a · 2023-10-21

**Soundness:** 3 good
**Presentation:** 4 excellent
**Contribution:** 3 good
**Rating:** 8
**Confidence:** 3

**Summary:**

It is well-established that neural networks are miscalibrated and overconfident on Out-Of-Distribution (OOD) data. At the same time, it is also known that classifiers tend to be less confident on OOD compared to In-Distribution data [1].
In this paper, the authors investigate the reasons why neural networks are less confident on OOD data, and propose the “reversion to the OCS” hypothesis as a possible explanation of this behavior. The Optimal Constant Solution (OCS) is a constant output to which neural networks converge when diverging from ID data. In the case of a classifier, the OCS corresponds to the uniform distribution over the target classes.

**Contributions**
In relation to the OCS hypothesis, the paper's contributions are the following:
- Empirical validation that the learned feature representations on OOD data have smaller norms (Fig. 4). This leads to the fact that the output of OOD is mostly driven by the input-independent components of the model (biases). Accordingly, the OCS is then mostly identified by the biases of the network (Fig. 4). Lastly, the role of the feature norms and network biases on OOD data was investigated theoretically in the restricted setting of Deep ReLU networks (Section 4.2).
- Empirical validation that as data deviates further from ID data, the output converges to the OCS (Fig. 3).
- In Section 5, the OCS hypothesis was leveraged as a tool for risk-sensitive decision-making. More in detail, we can define a classifier with a rejection option such that the OCS corresponds to preferring rejection to guessing one class. In this way, following the “reversion to the OCS” hypothesis, the further the classifier with rejection is from ID data, the more is prone to prefer the rejection option.

**Limitations**
The authors also addressed the limitations of the OCS hypothesis in the appendix, by showing that the hypothesis does not hold for adversarial examples (Fig. 8) and for specific types of noise (Fig. 9).

**References**
1. Hendrycks, D., & Gimpel, K. (2016). A baseline for detecting misclassified and out-of-distribution examples in neural networks.

**Strengths:**

- I found the paper very well written and clear. The experiments are all well presented and detailed.
- A lot of research has been recently devoted to addressing the issue of detecting overconfident OOD data and quantifying uncertainty in neural networks. This paper shows a result, that, in my opinion, might have been overlooked by the existing literature: while some OOD inputs might have high confidence, the more an input is far from ID data, the more the output *should* converge to some constant value.
- Existing OOD detectors, among the many, have exploited the norm of the learned features [1,2] and the confidence of the output [3] to detect OODs. This paper connects the two methodologies through the OCS hypothesis, by showing that OOD are less confident and detectable thanks to the lower feature norms and the reliance to the bias terms.

**References**
1. Sun, Yiyou, et al. "Out-of-distribution detection with deep nearest neighbors." International Conference on Machine Learning. PMLR, 2022.
2. Tack, Jihoon, et al. "Csi: Novelty detection via contrastive learning on distributionally shifted instances." Advances in neural information processing systems 33 (2020): 11839-11852.
3. Liang, S., Li, Y., & Srikant, R. (2017). Enhancing the reliability of out-of-distribution image detection in neural networks.

**Weaknesses:**

**Weaknesses**
- Lack of references: the norm of the learned features was already known to be a discriminant characteristic between ID and OOD data (see, e.g., [1] and [2] cited above).
- Lack of discussion on when the OCS does not hold. Although I understand that the results on the OCS hypothesis are still preliminary, I would like some additional remarks on when it might fail (as you showed in the appendix, but also as shown in works such as [1]). I suggest to briefly address this aspect in the introduction (see also Questions).
- I don't understand the conclusions drawn from the decision-making scenario. Ideas contained in the Hendrycks baselines paper ([1] in the summary) implicitly assume the OCS: if we choose an abstain threshold on the output confidence of a standard classifier, we already expect a non-zero abstain ratio on OOD data.  In general, anomaly detectors on neural networks based on the output confidence are based on these ideas. Could you explain better the intuitions provided by section 5 on exploiting the OCS? How does the OCS hypothesis possibly change the way in which we define safer AI models with rejection options, compared to simple threshold-based ones?

**Typos**:
- Page 1: "Therefore, the our hypothesis..."

**Concluding remarks**
Overall, I liked the paper thanks to its clarity and valid contributions, but I have some doubts on section 5 and the connections to related works. My final rating will depend on the other reviewer's comments and the authors' response.

**References**
1. Hein, Matthias, Maksym Andriushchenko, and Julian Bitterwolf. "Why relu networks yield high-confidence predictions far away from the training data and how to mitigate the problem." Proceedings of the IEEE/CVF Conference on Computer Vision and Pattern Recognition. 2019.

**Questions:**

- Do you see the OCS hypothesis as an "average" behavior of neural networks on OODs? In other works, do you expect that high-confidence OODs have to be constructed (e.g. adversarial examples), while the other ones will mostly respect the OCS hypothesis?
- In the OCS hypothesis, you clearly stated the assumption that OODs are high-dimensional. Is the OCS not valid on low-dimensional toy examples?
- In Figure 6, the OfficeHome dataset is the only one that has lower abstain ratio for the oracle compared to the predicted rewards. Is this because of the high number of target classes, compared to the other dataset, that makes it easier to achieve low confidence outputs?

---

> ### Author Response · Authors · 2023-11-18
> **Response to Reviewer 5N2a**
>
> We thank the reviewer for the insightful feedback and questions. Below we will address each of the weaknesses and questions.
>
> > I don't understand the conclusions drawn from the decision-making scenario. Ideas contained in the Hendrycks baselines paper ([1] in the summary) implicitly assume the OCS: if we choose an abstain threshold on the output confidence of a standard classifier, we already expect a non-zero abstain ratio on OOD data. … How does the OCS hypothesis possibly change the way in which we define safer AI models with rejection options, compared to simple threshold-based ones?
>
> We acknowledge that one limitation of our current study is that our proposed method may not improve on SOTA results in selective classification. Our main goal for section 5 is to illustrate how the choice of loss function could influence the performance of learned policies for solving general decision-making problems with OOD inputs, using selective classification as a simple example. Due to the single step nature of selective classification, there indeed exist simple “hard-coded” methods such as using an abstain threshold on the output confidence of a standard classifier whose performance might match or even surpass that of our proposed method. However, such “hard-coded” techniques do not exist for more complex decision-making problems involving multi-step reasoning. For instance, consider the problem of long-form question answering using large language models. When queried with an OOD question, we would like the model to generate more cautious responses such as “I don’t know”, or “The answer might be .. “ over highly confident but incorrect responses such as “The answer is [wrong answer]”. However, it would be challenging to eliciting this kind of linguistically calibrated behavior using “hard-coded” thresholding methods, as it requires reasoning about the uncertainty of a sequence of tokens, and some tokens may be more significant than others (e.g. “The answer is ..”/ “It is..” vs “Japan”/ “Germany”). Our work suggests that fine tuning with other forms of loss functions whose OCS corresponds to linguistically uncertain answers (e.g. reinforcement learning) may offer a promising and scalable solution to this problem. While we have not yet had a chance to fully explore this direction, we are very excited about the general principle of aligning the OCS of the loss function with cautious decisions, and we hope that our selective classification example can offer a first step towards illustrating its usage.
>
> > Lack of references: the norm of the learned features was already known to be a discriminant characteristic between ID and OOD data (see, e.g., [1] and [2] cited above).
>
> We thank the reviewer for pointing these prior works out to us; we have added them to the related works of our updated paper.
>
> > Lack of discussion on when the OCS does not hold. Although I understand that the results on - the OCS hypothesis are still preliminary, I would like some additional remarks on when it might fail (as you showed in the appendix, but also as shown in works such as [1]). I suggest to briefly address this aspect in the introduction (see also Questions).
>
> We thank the reviewer for this suggestion; we have added some discussion about where “reversion to the OCS” does not hold in the paper, including in the introduction.
>
> > Do you see the OCS hypothesis as an "average" behavior of neural networks on OODs? In other works, do you expect that high-confidence OODs have to be constructed (e.g. adversarial examples), while the other ones will mostly respect the OCS hypothesis?
>
> Yes, this is exactly our intuition. In our experiments, we found “reversion to the OCS” to appear in almost all the non-adversarial OOD datasets that we tried, but not for adversarial datasets.
>
> > In the OCS hypothesis, you clearly stated the assumption that OODs are high-dimensional. Is the OCS not valid on low-dimensional toy examples?
>
> In our experience, we have not found the “reversion to the OCS” phenomenon to hold on low-dimensional examples. For instance, consider this simple sin prediction example: https://ibb.co/Zf70F1K. Rather than reverting to the OCS (0), the model predictions moved farther away from the OCS as inputs became more OOD.
>
> > In Figure 6, the OfficeHome dataset is the only one that has lower abstain ratio for the oracle compared to the predicted rewards. Is this because of the high number of target classes, compared to the other dataset, that makes it easier to achieve low confidence outputs?
>
> Indeed, we have observed that MSE models sometimes tend to be a bit more prone to underfitting than CE models, especially when the number of classes are high, which may explain why they tend to predict low(er) confidence outputs even on the training distribution.

---

> ### Comment · Reviewer_5N2a · 2023-11-22
>
> I thank the authors for clarifying all my doubts.
>
> > We acknowledge that [...]
>
> Your intuition is indeed significant, and considering the extra information provided, I recognize its relevance.
>
> > In our experience, we have not found the “reversion to the OCS” phenomenon to hold on low-dimensional examples. [...]
>
> I think that also this point would have deserved some comments (at least in the appendix) as a limitation/future work on the OCS, together with adversarial examples.
>
> Given the discussion with the other reviewers, and your response, I confirm my rating. Overall, I enjoyed reading the paper, since it provides many interesting perspectives on OOD to think about.

---

### Official Review · Reviewer_Spup · 2023-10-23

**Soundness:** 2 fair
**Presentation:** 3 good
**Contribution:** 3 good
**Rating:** 6
**Confidence:** 4

**Summary:**

This paper focuses on the extrapolation of neural networks. It formulates a hypothesis that neural networks tend to move towards a constant value as the distribution shift on the input data increases. The first empirical observation exhibits that the norm of the feature representations decreases when the data are OOD and not sampled from ID. Then, the paper argues that this translates into the input-independent parts of the network (e.g. bias vectors) to dominate the output representation, thus explaining the more uniform prediction. The paper further argues that this constant value is close to the optimal constant solution (OCS). Then the paper focuses on why the OOD predictions have this tendency to move towards the OCS; the paper associates this behavior with the low-rank assumption on the learnt subspaces. The paper conducts a series of experiments with standard neural networks (e.g. resnet) to verify this hypothesis empirically. An application in high-risk decision making is also presented to exhibit the usefulness of this observation.

**Strengths:**

+ The paper is well-written, while understanding and improving the OOD of existing networks is important.

+ I personally like sec. A in the appendix, where the paper demonstrates some cases that have an unexpected performance with respect to their hypothesis.

+ The hypothesis seems new to me, while there are empirical results to support the hypothesis.

**Weaknesses:**

- Some of the training details are opaque in the main paper, which might lead into a simpler explanation over the observed empirical performance. For instance, could the learning algorithm or the data augmentation or the normalization impact this hypothesis?

- I am skeptical about the hypothesis formed in the following sense: even if we assume a zero input, most modern networks rely on a normalization scheme, e.g. batch or layer normalization. Then, in a trained network, the “centering” provided by the learnt means and variances of the network will not result in a zero-mean representation for the next layers. As such, I am wondering how the normalization plays into the formed hypothesis.

**Questions:**

Beyond the weaknesses above, the following questions come to mind about this submission:

- In fig. 4, I am wondering what the norm is for ID data; is it the case that the norm is also decreasing on those networks as well?

- In theorem 4.1 it mentions a “shallow” network with L’ layers. Is this a contradiction?

- Even though not strictly mandatory, I am wondering whether structures that differ from the feedforward structure of MLPs differ in the solution they find. For instance, graph neural networks or multi-path networks. In other words, I am not sure whether the proved extrapolation properties (see Xu et al 2021 and Wu et al 2022 that are already cited) affect the formed hypothesis.

---

> ### Author Response · Authors · 2023-11-15
> **Response to Reviewer Spup (1/2)**
>
> We thank the reviewer for the feedback and questions. Below, we will present additional experiments analyzing the effects of normalization, as well as answering the reviewer’s questions with regards to training details, data augmentation, and details about Fig 4 and Thm 4.1. We believe these clarifications address the main weaknesses stated in the review, but please let us know if any further concerns remain, we would like to do our best to address these!
>
> > I am wondering how the normalization plays into the formed hypothesis.
>
> To better understand the role of normalization, we trained a neural network with no normalizations (NN), batch norm (BN), and layer norm (LM) using cross entropy on the MNIST dataset (we visualize the specific neural network architectures used here: https://ibb.co/c85zgjh).  We then evaluated these models on OOD inputs with increasing levels of distribution shift (rotations of 15, 30, 45, 60, 75 degrees), and measured the distance between the models’ predictions from the OCS, which we visualize here: https://ibb.co/DMQx6D9). We can see that “reversion to the OCS” holds for all three models.
>
> Next we analyze the representations produced by each of the models when presented with OOD inputs. We conduct the same experiments as presented in Fig. 4 of our submission on all three models (the no normalization model is the same as the one in the paper), which we visualize here: https://ibb.co/qYghdzS). We can see that the relative norm of the representations tend to decrease as distribution shift increases in the later layers of the model for all three models (left). Furthermore, we can see similar behaviors for all three models with regards to the proportion of network features which lie within the span of the following linear layer (middle), and the similarity between the accumulation of model biases to the OCS (right).
>
> While normalizations may play a role in enhancing “reversion to the OCS”, and additional work is needed to more precisely characterize the nature of its effect, the experiments we presented here showed that normalization alone does not explain the “reversion to the OCS” phenomenon, because it still exists for models without normalizations. Furthermore, there is evidence in both models with or without normalizations for our hypothesis on the explanation behind the “reversion to the OCS” phenomenon
>
>
> > Some of the training details are opaque in the main paper, which might lead into a simpler explanation over the observed empirical performance. Could the learning algorithm or the data augmentation or the normalization impact this hypothesis?
>
>
> Having addressed the impact of normalization in the previous section, we will focus on the learning algorithm and the data augmentation. With regards to the learning algorithm, we restrict our observations to empirical risk minimization of deep neural networks using gradient descent, which we specify in Section 3. With regards to data augmentation, our experiments show the “reversion to the OCS” phenomenon on models both trained with (ImageNet, OfficeHome, BREEDS) and without (MNIST, SkinLesionPixels, UTKFace) data augmentation. Thus, while data augmentation may have an effect on the extent of “reversion to the OCS”, we believe “reversion to the OCS” cannot solely be explained by data augmentation.
>
> With regards to the training details, we have updated the experiments section of Section 4 to more explicitly mention that the MNIST analysis experiments use models trained without data augmentation and without normalizations, while the CIFAR10 analysis experiments use models with data augmentations and with batch norm. Further training details for every model we trained in the paper can be found in Section B of the Appendix, including the specific neural network architecture, optimizer, optimizer parameters, and data preprocessing. Please let us know if there are any other training details you would like to see that we did not include, and we would be happy to add them.
>
>
> > In fig. 4, I am wondering what the norm is for ID data; is it the case that the norm is also decreasing on those networks as well?
>
>
> In the left plot in Figure 4, the quantity we are plotting on the y-axis is the relative norm between OOD representations and ID representations (average norm of OOD representations divided by the average norm of ID representations). Thus, the ID data will correspond to a value of 1 for each layer.

---

> > ### Author Response · Authors · 2023-11-15
> > **Response to Reviewer Spup (2/2)**
> >
> > > In theorem 4.1 it mentions a “shallow” network with L’ layers. Is this a contradiction?
> >
> >
> > Thank you for pointing out the confusing wording of Theorem 4.1, we have updated the paper such that the theorem is stated in more clear terms.
> >
> > Note that the networks we consider can indeed have a large number of layers $L$. By “shallow”, what we meant was that if there exists a relatively shallower network (with number of layers $L' < L$) that can fit the data with some margin $\gamma$, then the network we train learns low rank weights, i.e. the learnt weights $\hat{W}$ imply the existence of low rank subspaces $\{A_i\}_{i=1}^L$ that satisfy the property (on OOD data) in Thm 4.1.

---

> ### Comment · Reviewer_Spup · 2023-11-18
> **Response**
>
> Dear authors,
>
> I am thankful for the provided response. Nevertheless, I am not convinced by the provided explanation on normalization. The response provides an empirical way with MNIST data. However, in MNIST it is possible to even learn a linear classifier (or directly classify in the input space) and obtain an accuracy > 90%. Therefore, I am not very convinced about the answer or the fact that there is no assumption about one of the core parts of modern networks (i.e. normalization).

---

> ### Author Response · Authors · 2023-11-21
> **Response to Reviewer Spup**
>
> To address the reviewer's concern, we include additional experiments training an AlexNet models with no normalization (NN), batch normalization (BN), and layer normalization (LN) on CIFAR10, and evaluating on CIFAR10-C datasets. We chose to use AlexNet instead of ResNet20 (which we used for CIFAR10 experiment in the paper), because ResNet20 was numerically unstable without batch norm. Our experiments, visualized here: https://ibb.co/Fmc98YQ, show consistent behavior with our MNIST analysis from the rebuttal as well as the ones presented in the paper.

---

> > ### Comment · Reviewer_Spup · 2023-11-21
> > **Is this newly added experiment in the paper? Where can I find the details?**
> >
> > I am thankful to the authors for including those experiments. Is this experiment described somewhere in the paper? Because I find a very vague description of the experiment in the response. If the experiments are already in the paper, I can find the details directly.

---

> > > ### Author Response · Authors · 2023-11-21
> > > **Response**
> > >
> > > Yes, the experiment is described in Section 4.1 of our paper. The main differences between the new results and the ones presented in the paper are that 1) we use AlexNet instead of ResNet20, and 2) we trained on 3 different versions of the model with no norm, batch norm, and layer norm.

---

> > > > ### Comment · Reviewer_Spup · 2023-11-22
> > > > **Why are the results with normalization excluded from the paper?**
> > > >
> > > > I just noticed that the results on the normalization are not included in the paper and I am curious what the reason behind this is.

---

> > > > > ### Author Response · Authors · 2023-11-23
> > > > > **Response**
> > > > >
> > > > > Apologies, we hadn't had a chance to update the paper yet in our last response. We have now updated the paper to include these experiments (see Sections 4.1 and Appendix C).

---

> > > > > > ### Comment · Reviewer_Spup · 2023-11-23
> > > > > > **Response**
> > > > > >
> > > > > > Thank you for the quick response. Even though I am not fully convinced about the normalization exploration, I am thankful to the authors for their effort and have thus increased my score to 6. I do believe there are some weaker points and we need (as a community) to be very careful before making strong claims about biases (or extrapolation in this case), but the paper does a decent job to address those.
> > > > > >
> > > > > > I hope the camera-ready version does explore further this phenomenon with the normalization and forms a hypothesis that accounts for normalization layers.

---

### Official Review · Reviewer_KwvU · 2023-10-31

**Soundness:** 3 good
**Presentation:** 3 good
**Contribution:** 2 fair
**Rating:** 6
**Confidence:** 3

**Summary:**

The paper aims to show that, contrary to the (somewhat) common belief, OOD inputs mostly lead to predictions closer to the average of training labels (more specifically, the optimal constant solution, OCS, minimizing the training loss without the inputs), as opposed to overconfident incorrect results reported in many previous works. The paper first demonstrates the strong (negative) correlation between the distance to OCS and the OOD score (estimated by separately trained low-capacity models) over 8 image & text datasets using ResNet, VGG and DistilBERT (Fig 3), then shows the reason is that OOD inputs tend to have smaller projected feature norms $||Wh||$ (Fig 4), which could be theoretically explained within homogeneous DNNs with ReLU activations (Thm 4.1). Finally the paper presents an OCS-based selective classification algorithm using MSE loss, and validates its performance (abstain ratio, reward) against the standard (CE) classification and an oracle (with access to evaluation data) on 4 CV datasets (Fig 6 & 7).

**Strengths:**

+ [Originality] The paper novelly reassesses DNN’s OOD behavior both empirically and theoretically (OCS, OOD score, Sec 4.2), and reports interesting results (dispute of the common belief, OCS-based selective classification).
+ [Quality] The paper is of sufficient quality in my opinion, with proper empirical (Fig 3 & 4) and theoretical (Thm 4.1, Prop 4.2) validations of the main claims, and experimental evidence of the proposed algorithm’s effectiveness (Fig 6 & 7, although can be further improved, see Weaknesses).
+ [Clarity] The paper is overall clear and easy to follow, although certain details, e.g. the construction of the $(x,a,r)$ dataset in Sec 5.1, can be further elaborated for better understandability.

**Weaknesses:**

- [Evaluation] While it’s understandable that this work doesn’t focus on achieving SOTA results, it’s still highly desirable to see how the proposed algorithm compares to existing selective classification (or OOD detection) baselines, and/or how they can be combined to further boost performance (discussion would be fine too).
- [Significance] While this paper is good in most aspects (as summarized in Strengths), its significance however is a bit insufficient in my opinion and can be substantially improved by addressing the following issues:
1) Evaluation as stated above. More evaluation can help strengthen the applicability of the paper.
2) The failure of the claim in some of the distribution shifts (adversarial perturbation and impulse noise in Appendix A) raises concerns about the generalizability of this work. What kinds of OOD shifts (extrapolations) are actually supported and not supported by this work? Is there a way to more formally and/or finely characterize them? More evaluation datasets as well as systematically generated OOD shifts e.g. [1] could be helpful.

[1] A Fine-Grained Analysis on Distribution Shift, ICLR, 2022.

**Questions:**

* In Fig 7, reward prediction seems to be noticeably better than standard classification on all CIFAR10 noises and particularly OfficeHome even at the training distribution (t), i.e. supposedly not OOD. Is this solely because of abstention (as shown in Fig 6), or does MSE loss somehow works better than CE loss in this case? More generally, how do their accuracies (instead of rewards) compare e.g. without considering the abstained samples?
* In Fig 4, the normalized norm in early layers for CIFAR10 (bottom left panel) seems to be systematically increasing with noise, contrary to later layers. Is this an expected behavior?
* Is there a particular reason to use -4 for the incorrect results? Does any number that brings the OCS below 0 (-3.5 in the paper) work?

---

> ### Author Response · Authors · 2023-11-18
> **Response to Reviewer KwvU**
>
> We thank the reviewer for the feedback and questions. Below, we will present additional experiments detailing the accuracy of the selective classification models, the behavior of early layers, and the performance of the selective classification models with a different reward function. We additionally discuss the relationship between our approach and the SOTA methods for selective classification, as well as characterizing when “reversion to the OCS” does and does not hold.
>
> > In Fig 7, reward prediction seems to be noticeably better than standard classification on all CIFAR10 noises and particularly OfficeHome even at the training distribution (t), i.e. supposedly not OOD. Is this solely because of abstention (as shown in Fig 6), or does MSE loss somehow works better than CE loss in this case? More generally, how do their accuracies (instead of rewards) compare e.g. without considering the abstained samples?
>
> We visualize the accuracy of each model on all the ID/OOD datasets here: https://ibb.co/nnsfvmJ. In general, the accuracy of the CE model is about the same or a bit higher than the MSE model. Thus, the reason the MSE model has higher reward than the BC model is because it is able to abstain better, not because it has higher accuracy.
>
> > In Fig 4, the normalized norm in early layers for CIFAR10 (bottom left panel) seems to be systematically increasing with noise, contrary to later layers. Is this an expected behavior?
>
> Not necessarily; we found the behavior of the early layers to change depending on the model and the evaluation dataset. For instance, evaluating the CIFAR10 models on Gaussian noise produces normalized norms which don't have this behavior in the early layers:  https://ibb.co/Nt9LrZR.
>
> > Is there a particular reason to use -4 for the incorrect results? Does any number that brings the OCS below 0 (-3.5 in the paper) work?
>
> Yes, any number that brings the OCS below 0 should work. For instance, we additionally include an experiment using a reward of -8 for incorrect predictions here: https://ibb.co/7CtMvJ7. Here, we can see that the difference between the reward prediction and imitation models is bigger than the one in the paper (with -4 incorrect prediction reward), because there is now a higher penalty for choosing an incorrect prediction.
>
> > While it’s understandable that this work doesn’t focus on achieving SOTA results, it’s still highly desirable to see how the proposed algorithm compares to existing selective classification (or OOD detection) baselines, and/or how they can be combined to further boost performance (discussion would be fine too).
>
> We don't believe that our method will be better than the SOTA approach in selective classification. However, insights from SOTA methods can indeed be combined with ours to potentially boost performance. For instance, [1] found that leveraging unlabeled data to train the model via an unsupervised loss can improve the performance in selective classification by a significant margin. While our method currently does not make use of unlabeled data, adding the use of unlabeled data during training may lead to significant improvements in our method as well. We hope to investigate this (and other techniques) further in future work.
>
> [1] Towards Better Selective Classification, ICLR 2023
>
> > The failure of the claim in some of the distribution shifts (adversarial perturbation and impulse noise in Appendix A) raises concerns about the generalizability of this work. What kinds of OOD shifts (extrapolations) are actually supported and not supported by this work? Is there a way to more formally and/or finely characterize them?
>
> Our intuition is that distribution shifts which cause the distribution of model representations to differ from those of training inputs will lead to “reversion to the OCS”, whereas those that don’t will not. In our analysis in 4.1, we found that naturally occurring or non-adversarial shifts tend to cause the representations to shift from the training representations. If we model non-adversarial shifts as adding “noise” to the directions of the representation vectors, then we would expect the overall distribution of representations to also shift, because the training representation distribution usually lies in a low dimensional space. In contrast, because adversarial inputs are optimized to maximize loss, they may be more likely to map to representations which align with the training representations. Indeed, we found “reversion to OCS” to hold for almost all the non-adversarial shifts we tried, but not adversarial shifts. Of course, the intuition we presented above is not rigorous, and more work is needed to characterize it in a more precise and rigorous manner. We hope to tackle this important question in future work.

---

### Official Review · Reviewer_DSRT · 2023-11-01

**Soundness:** 3 good
**Presentation:** 3 good
**Contribution:** 4 excellent
**Rating:** 8
**Confidence:** 4

**Summary:**

The paper proposes the "reversion to OCS" hypothesis: Neural network predictions often tend towards a constant value as input data becomes increasingly OOD; that constant value closely approximates the optimal constant solution (OCS), which is the prediction that minimizes the average loss over the training data without observing the input. This hypothesis is verified by empirical results on a wide range of datasets and architectures with different input modalities and loss functions. Moreover, the paper also provides theoretical results to explain the observed behavior. Specifically, the feature norm of each neural layer can drop easily with OOD inputs, which shows why the model's output converges to the OCS when the input becomes more OOD. Finally, the authors leverage this insight to enable risk-sensitive decision-making.

**Strengths:**

* Finding of the paper is interesting.

* Paper writing is careful and clear.

* The paper includes detailed evidence, both empirically and theoretically, for their claims.

**Weaknesses:**

There is no significant weakness.

**Questions:**

1. Do behaviors of $W_i \phi_I(x)$ mentioned in Section 4.1 remain on realistic datasets, such as ImageNet and Amazon?

2. From the insight of the paper, can we say that methods that try to improve models’ performances on OOD inputs are actually moving the models’ outputs away from the OCS?

---

> ### Author Response · Authors · 2023-11-18
> **Response to Reviewer DSRT**
>
> We thank the reviewer for the review and the questions. We will address the questions below.
>
> > Do behaviors of mentioned in Section 4.1 remain on realistic datasets, such as ImageNet and Amazon?
>
> To address this question, we conducted new analysis experiments on ImageNet (evaluated on ImageNet-R(endition) and ImageNet-Sketch), and our results show similar trends as those in the paper. Our results can be found here: https://ibb.co/x2gV4pY
>
>
> > From the insight of the paper, can we say that methods that try to improve models’ performances on OOD inputs are actually moving the models’ outputs away from the OCS?
>
> Not necessarily. For cross entropy, a model’s performance can be improved by 1) changing a low confidence prediction to a highly confident and correct prediction, 2) changing a highly confident incorrect prediction to a highly confident correct prediction, 3) changing a highly confident but incorrect prediction to a low confidence prediction. Here, 1) will cause the model outputs to move away from the OCS, 2) will cause the model outputs to remain about the same distance to the OCS, and 3) will cause the model outputs to move towards the OCS. Thus, it is possible to improve the performance of a model on OOD inputs while moving the model’s outputs more towards the OCS.

---

### Meta-Review · Area_Chair_ypSX · 2023-12-06

**Metareview:**

This paper questions the wisdom that neural networks extrapolate wildly for out-of-distribution (OOD) inputs. Instead, this paper presents empirical and theoretical evidence that neural networks revert to predicting the marginal distribution over y (the “optimal constant” solution) as data becomes more OOD. Overall, the empirical and theoretical evidence in this paper is thorough and convincing, although there are some lingering questions about how normalization layers affect convergence to the optimal constant solution. This paper confirms observations made in previous papers on uncertainty quantification and OOD detection. Crucially, it illustrates a subtle but important point that counters existing intuitions, and does so with compelling evidence. For this reason, it should be accepted into ICLR 2024.

**Justification For Why Not Higher Score:**

While this paper provides compelling theoretical and empirical evidence, it - to some extent - confirms findings that were already present in the literature. For example, [1] already demonstrates that low confidence predictions (i.e. predictions close to the mean) are likely to be OOD. Therefore, I would argue that this contributions is not "groundbreaking," but it nevertheless provides significant and meaningful evidence that will be a contribution to the research community.

[1] Hendrycks, D., & Gimpel, K. (2016). A baseline for detecting misclassified and out-of-distribution examples in neural networks.

**Justification For Why Not Lower Score:**

The experiments and theory provided by this paper are solid and well grounded. This paper tackles a problem that is significant and timely, and provides concrete evidence that counters commonly-held intuitions by the community.

---

### Decision · Program_Chairs · 2024-01-16

Accept (poster)